# A seasonal analysis of aerosol $NO_3^-$ sources and $NO_x$ oxidation pathways in the Southern Ocean marine boundary layer

**Jessica M. Burger**[1]**, Emily Joyce**[2,3]**, Meredith G. Hastings**[2,3]**, Kurt A. M. Spence**[1]**, and Katye E. Altieri**[1]

[1]Department of Oceanography, University of Cape Town, Rondebosch, 7701, South Africa
[2]Department of Earth, Environmental and Planetary Science, Brown University, Providence, RI 02906, USA
[3]Institute at Brown for Environment and Society, Brown University, Providence, RI 02906, USA

**Correspondence:** Jessica M. Burger (brgjes006@uct.ac.za)

**Abstract.** Nitrogen oxides, collectively referred to as $NO_x$ ($NO + NO_2$), are an important component of atmospheric chemistry involved in the production and destruction of various oxidants that contribute to the oxidative capacity of the troposphere. The primary sink for $NO_x$ is atmospheric nitrate, which has an influence on climate and the biogeochemical cycling of reactive nitrogen. $NO_x$ sources and $NO_x$-to-$NO_3^-$ formation pathways remain poorly constrained in the remote marine boundary layer of the Southern Ocean, particularly outside of the more frequently sampled summer months. This study presents seasonally resolved measurements of the isotopic composition ($\delta^{15}N$, $\delta^{18}O$, and $\Delta^{17}O$) of atmospheric nitrate in coarse-mode ($> 1\,\mu m$) aerosols, collected between South Africa and the sea ice edge in summer, winter, and spring. Similar latitudinal trends in $\delta^{15}N-NO_3^-$ were observed in summer and spring, suggesting similar $NO_x$ sources. Based on $\delta^{15}N-NO_3^-$, the main $NO_x$ sources were likely a combination of lightning, biomass burning, and/or soil emissions at the low latitudes, as well as oceanic alkyl nitrates and snowpack emissions from continental Antarctica or the sea ice at the mid-latitudes and high latitudes, respectively. Snowpack emissions associated with photolysis were derived from both the Antarctic snowpack and snow on sea ice. A combination of natural $NO_x$ sources, likely transported from the lower-latitude Atlantic, contribute to the background-level $NO_3^-$ observed in winter, with the potential for a stratospheric $NO_3^-$ source evidenced by one sample of Antarctic origin. Greater values of $\delta^{18}O-NO_3^-$ in spring and winter compared to summer suggest an increased influence of oxidation pathways that incorporate oxygen atoms from $O_3$ into the end product $NO_3^-$ (i.e. $N_2O_5$, DMS, and halogen oxides (XO)). Significant linear relationships between $\delta^{18}O$ and $\Delta^{17}O$ suggest isotopic mixing between $H_2O_{(v)}$ and $O_3$ in winter and isotopic mixing between $H_2O_{(v)}$ and $O_3$/XO in spring. The onset of sunlight in spring, coupled with large sea ice extent, can activate chlorine chemistry with the potential to increase peroxy radical concentrations, contributing to oxidant chemistry in the marine boundary layer. As a result, isotopic mixing with an additional third end-member (atmospheric $O_2$) occurs in spring.

## 1 Introduction

The atmosphere of the Southern Ocean is geographically remote from major anthropogenic influences. Although there is evidence of microplastics and at times long-range transport of anthropogenic pollution (Jacobi et al., 2000; Obbard, 2018), the Southern Ocean marine boundary layer (MBL) is one of the few regions dominated by natural sources, and as such at times it can serve as a proxy for the pre-industrial atmosphere. The pre-industrial atmosphere is used as a baseline for comparing the magnitude of anthropogenic impacts on climate (e.g. Haywood and Boucher, 2000; Hamilton et al., 2014; Schmale et al., 2019).

Nitrogen oxides (NO$_x$ = NO + NO$_2$) are an important part of biogeochemical cycling and influence the oxidative capacity of the troposphere as they are involved in the production and destruction of ozone and hydroxyl radicals (Lawrence and Crutzen, 1999; Finlayson-Pitts and Pitts, 2000). The primary sink for NO$_x$ is atmospheric nitrate (NO$_3^-$), which impacts both air quality and climate by influencing particulate matter load and Earth's radiative heat budget (IPCC, 2013; Park and Kim, 2005).

The logistical difficulties of measurement campaigns to the remote Southern Ocean, particularly in winter, have resulted in a lack of observational data from this region, including data on NO$_x$ sources and sinks (Paton-Walsh et al., 2022). Consequently, the seasonality of NO$_x$ cycling remains poorly constrained in the Southern Ocean MBL. Globally, fossil fuel combustion is the primary NO$_x$ source (van der A et al., 2008), far exceeding natural emissions like biomass burning (Finlayson-Pitts and Pitts, 2000), soil processes (Davidson and Kingerlee, 1997), and lightning (Schumann and Huntrieser, 2007). However, regional budgets of NO$_x$ sources can have a variety of anthropogenic and natural contributors. In the summertime Southern Ocean MBL, natural NO$_x$ sources are the main contributors to atmospheric NO$_3^-$ formation (Morin et al., 2009; Burger et al., 2022a). Along the South African coastline, these natural NO$_x$ sources include a combination of lightning, biomass burning, and soil emissions (Morin et al., 2009). In coastal Antarctica, or near the marginal ice zone, NO$_x$ emitted from snow cover serves as the main precursor to atmospheric NO$_3^-$ (Savarino et al., 2007; Morin et al., 2009; Shi et al., 2021; Burger et al., 2022a). Over the mid-latitude region of the Southern Ocean, sea surface emissions of a group of nitrogen gases referred to as alkyl nitrates (RONO$_2$) have recently been proposed as a NO$_x$ source leading to NO$_3^-$ formation in the MBL (Fisher et al., 2018; Burger et al., 2022a). During winter, NO$_x$ sources to the Antarctic troposphere primarily include long-range-transported peroxyacetyl nitrate (PAN) and stratospheric inputs (Savarino et al., 2007; Lee et al., 2014; Walters et al., 2019). To our knowledge, however, there are no observational data regarding NO$_x$ sources from the Southern Ocean MBL during winter and few observations in spring.

In addition to there being multiple NO$_x$ sources across the Southern Ocean MBL, several different oxidation pathways can be responsible for NO$_x$-to-NO$_3^-$ conversion, varying with chemistry and time of day (Savarino et al., 2007). Once emitted, NO is rapidly oxidized by ozone (O$_3$) (Reaction R1), peroxy radicals (RO$_2$ or HO$_2$) (Reaction R2), and/or halogen oxides (XO, where X = Br, Cl, or I) (Reaction R3) to NO$_2$.

$$NO + O_3 \rightarrow NO_2 + O_2 \tag{R1}$$

$$NO + RO_2 \ (or \ HO_2) \rightarrow NO_2 + RO \ (or \ OH) \tag{R2}$$

$$NO + XO \rightarrow NO_2 + X \tag{R3}$$

$$NO_2 + O_2 + hv \rightarrow NO + O_3 \tag{R4}$$

Under sunlit conditions, NO$_2$ is readily photolysed to regenerate NO and O$_3$ (Reaction R4). The recycling of NO$_x$ between NO and NO$_2$ happens much faster than NO$_x$ oxidation to NO$_3^-$ during the day (Michalski et al., 2003). On a global scale, NO is primarily oxidized to NO$_2$ by O$_3$, followed by HO$_2$ and RO$_2$, while NO-to-NO$_2$ oxidation via XO is relatively minor (Alexander et al., 2020).

During summer in the Southern Ocean MBL, NO$_2$ is subsequently oxidized primarily by hydroxyl radicals (OH) to form HNO$_3$ (Reaction R5).

$$NO_2 + OH + M \rightarrow HNO_3 + M \tag{R5}$$

In winter, under dark conditions, when the photolytic production of OH stops, NO$_2$ is oxidized primarily by O$_3$ to form nitrate radicals (NO$_3$) (Reaction R6). NO$_3$ can then react with NO$_2$ to form dinitrogen pentoxide (N$_2$O$_5$) followed by hydrolysis on a wet particle surface to form HNO$_3$ (Reactions R7–R8).

$$NO_2 + O_3 \rightarrow NO_3 + O_2 \tag{R6}$$

$$NO_3 + NO_2 + M \rightleftharpoons N_2O_{5(g)} + M \tag{R7}$$

$$N_2O_{5(g)} + H_2O_{(l)} + surface \rightarrow 2HNO_{3(aq)} \tag{R8}$$

Alternatively, HNO$_3$ can be formed by the reaction of NO$_3$ with hydrocarbons (HC) (e.g. dimethylsulfide, DMS) (Reaction R9).

$$NO_3 + HC \ or \ DMS \rightarrow HNO_3 + products \tag{R9}$$

Lastly, halogen chemistry may result in NO$_3^-$ formation via the production and subsequent hydrolysis of halogen nitrates (Reactions R10–R11), as has been suggested for coastal Antarctica in summer (Baugitte et al., 2012).

$$XO + NO_2 \rightarrow XNO_3 \tag{R10}$$

$$XNO_3 + H_2O_{(l)} + surface \rightarrow HNO_{3(aq)} + HOX \tag{R11}$$

The nitrogen (N) and oxygen (O) isotopic composition of atmospheric NO$_3^-$ provides information regarding NO$_x$ sources and NO$_3^-$ formation pathways (i.e. NO oxidation to NO$_2$ and NO$_2$ oxidation to NO$_3^-$). This technique has been applied in polluted (Elliot et al., 2007; Zong et al., 2017), open-ocean (Hastings et al., 2003; Altieri et al., 2013; Kamezaki et al., 2019; Burger et al., 2022a), and polar environments (Walters et al., 2019). Stable isotope ratios are reported as the ratio of the heavy to light isotopologues of a sample relative to the constant isotopic ratio of a reference

standard using delta ($\delta$) notation in units of "per mil" (‰) following Eq. (1):

$$\delta = \left((R_{\text{sample}}/R_{\text{standard}}) - 1\right) \times 1000, \tag{1}$$

where $R$ represents the ratio of $^{15}$N / $^{14}$N, $^{18}$O / $^{16}$O, or $^{17}$O / $^{16}$O in the sample and in the reference standard, respectively. The reference for O is Vienna Standard Mean Ocean Water (VSMOW), and for N it is atmospheric N$_2$ (Bölhke et al., 2003).

The N isotopic composition of atmospheric NO$_3^-$ ($\delta^{15}$N–NO$_3^-$) largely reflects the $\delta^{15}$N of different precursor NO$_x$ emissions (e.g. Elliott et al., 2019, and references therein) but can be influenced by isotopic fractionation during NO$_x$ cycling and NO$_x$-to-NO$_3^-$ conversion (Walters and Michalski 2015; Walters et al., 2016; Li et al., 2021). $\delta^{15}$N–NO$_3^-$ is therefore useful for constraining NO$_x$ sources. For example, biomass burning may produce NO$_x$ with a $\delta^{15}$N range of $-7$‰ to $12$‰ (Fibiger and Hastings, 2016), while soils lead to NO$_x$ emissions with relatively low $\delta^{15}$N signatures ($-44.2$‰ to $-14.0$‰; Miller et al., 2018). More relevant to the remote Southern Ocean is lightning NO$_x$, which has a $\delta^{15}$N signature of approximately $0$‰ (Hoering, 1957). This is distinct from the snowpack NO$_x$ source, which typically has a very low $\delta^{15}$N signature (Berhanu et al., 2014, 2015) on the order of $-50$‰ to $-20$‰ (Wagenbach et al., 1998; Winton et al., 2020), depending on the degree of snowpack NO$_3^-$ $^{15}$N enrichment (Shi et al., 2018). Savarino et al. (2007) derived an Antarctic stratospheric NO$_x$ source signature of $19 \pm 3$‰. Additionally, Burger et al. (2022a) estimated the $\delta^{15}$N signature of NO$_x$ produced by surface ocean RONO$_2$ emissions over the mid-latitude Southern Ocean to be $-21.8 \pm 7.6$‰.

The O isotopic composition of atmospheric NO$_3^-$ ($\delta^{18}$O–NO$_3^-$ and $\Delta^{17}$O–NO$_3^-$) reflects the oxidants responsible for NO$_3^-$ formation, as atmospheric oxidants have distinct O isotope signatures (Michalski et al., 2012). $\delta^{18}$O–NO$_3^-$ and $\Delta^{17}$O–NO$_3^-$ are thus useful for identifying pathways of NO$_3^-$ production (Michalski et al., 2003; Hastings et al., 2003; Alexander et al., 2020). O$_3$ possesses a distinctively large $^{17}$O excess as a result of non-mass-dependent isotope fractionation. This $^{17}$O excess is expressed as $\Delta^{17}$O $= \delta^{17}$O $- 0.52 \times \delta^{18}$O (Berhanu et al., 2012). Non-mass-dependent fractionation occurs in the troposphere and is thought to originate from asymmetric molecules of excited ozone (O$_3^*$) that lose excess energy via stabilization to the product O$_3$ (Heidenreich and Thiemens, 1986; Ireland et al., 2020). As a result, O$_3$ possesses a uniquely high terminal $\Delta^{17}$O $= 39.2 \pm 2$‰ (Vicars and Savarino, 2014) that can be transferred to NO$_3^-$ during oxidation reactions between NO$_x$ and O$_3$ (Thiemens, 2006; Savarino et al., 2008; Michalski and Bhattacharya, 2009) or NO$_x$ and other oxidants like XO where the oxygen atom originated from O$_3$ (Savarino et al., 2016). $\Delta^{17}$O–NO$_3^-$ therefore can serve as a proxy for the in-

fluence of O$_3$ and/or XO during NO$_3^-$ formation (Berhanu et al., 2012).

O$_3$ also has a uniquely high terminal $\delta^{18}$O $= 126.3 \pm 11.9$‰ (Vicars and Savarino, 2014) compared to other oxidants that have a $\Delta^{17}$O of $0$‰ and much lower $\delta^{18}$O signatures (Michalski et al., 2003, 2012). For example, atmospheric O$_2$ has a $\delta^{18}$O of $23.9$‰, and the $\delta^{18}$O of OH and H$_2$O are CE1 negative TS1 (Michalski et al., 2012). As such, a higher $\delta^{18}$O or $\Delta^{17}$O for atmospheric NO$_3^-$ reflects the increased influence of O$_3$ and/or XO on NO$_3^-$ formation, while a lower $\delta^{18}$O or $\Delta^{17}$O occurs when there is an increased contribution from other oxidants (Hastings et al., 2003; Fang et al., 2011; Altieri et al., 2013). Oxidation by peroxy radicals would also result in a lower $\delta^{18}$O and $\Delta^{17}$O signature for atmospheric nitrate because the O atom in peroxy radicals derives from atmospheric O$_2$.

Antarctic tropospheric oxidation chemistry has been well characterized using $\Delta^{17}$O at coastal (Savarino et al., 2007; Ishino et al., 2017) and interior Antarctic sites (Frey et al., 2009; Savarino et al., 2016; Walters et al., 2019). A distinct seasonal cycle in $\Delta^{17}$O–NO$_3^-$ is generally observed whereby a higher relative contribution from O$_3$ oxidation and/or stratospheric input occurs during winter, and more HO$_x$ + RO$_x$ oxidation occurs during summer. The Atlantic Southern Ocean is less constrained in terms of oxidation chemistry, with some evidence that the atmospheric oxidant budget is poorly understood in unpolluted low-NO$_x$ environments (Hosaynali Beygi et al., 2011).

This study presents the first seasonally resolved dataset of coarse-mode ($> 1$ µm) atmospheric NO$_3^-$ concentration and isotopic composition from the Atlantic Southern Ocean MBL. Using air mass back trajectories and observed aerosol $\delta^{15}$N–NO$_3^-$, $\delta^{18}$O–NO$_3^-$, and $\Delta^{17}$O–NO$_3^-$ between Cape Town, South Africa, and the marginal ice zone, this work aims to identify how the main sources and formation pathways of NO$_3^-$ vary over the remote Southern Ocean from winter through spring and summer.

## 2 Methods

### 2.1 Sample collection

Samples were collected on board the research vessel (R/V) *SA Agulhas II* during three voyages to and from the marginal ice zone in summer (7 to 21 December 2018 and 27 February to 15 March 2019), winter (19 July to 12 August 2019), and spring (13 October to 19 November 2019) (Fig. 1). The summer samples presented here are the same as those in Burger et al. (2022a). The winter and spring samples were collected and analysed as in Burger et al. (2022a), with any methodological modifications noted below. Briefly, all voyages departed from Cape Town (33.9° S, 18.4° E) and sailed southward along the Good Hope transect (0° E) until reaching Penguin Bukta (71.4° S, 2.5° W) in summer and the northern extent of the sea ice in winter (approximately 58.1° S)

and spring (approximately 59.3° S). The ship then returned to Cape Town, sailing north via the Good Hope transect, with a deviation to South Georgia in the summer. In spring an additional ice edge transect was conducted during which the ship sailed from 0 to approximately 22° E and back before returning to Cape Town.

Size-segregated aerosols were collected on the ninth floor above the bridge (approximately 20 m above sea level), using a high-volume air sampler (HV-AS; Tisch Environmental). Air was pumped at an average flow rate of 1.3 m$^3$ min$^{-1}$ through a five-stage cascade impactor (TE-235; Tisch Environmental), loaded with pre-combusted (400 °C for 4 h) glass fibre filters. Given that aerosol nitrate in the MBL is predominantly present in the coarse mode ($> 1$ µm), only filter stages 1 through 4 were analysed. The aerodynamical diameters of particles captured by filter stages 1, 2, 3, and 4 are $> 7$, 3 to 7, 1.5 to 3, and 1 to 1.5 µm, respectively.

A sector collector was used to restrict HV-AS activity to avoid contamination of the filters with ship stack emissions (Campbell Scientific Africa). The HV-AS only operated if the winds were blowing at an angle less than 120° or greater than 240° from the bow of the ship during winter and less than 75° or greater than 190° from the bow of the ship during spring. These criteria were altered based on the dominant wind direction during each voyage to ensure sufficient sample collection while avoiding contamination. In addition to wind direction, the wind speed had to exceed 0 m s$^{-1}$ for 10 min for the HV-AS to begin sampling. Filters were removed from the cascade impactor inside a laminar-flow cabinet (Air Science), placed in individual zip-sealed bags and stored at $-20$ °C until analysis.

An attempt was made to ensure that there was at least 24 h of in-sector sampling before removing filters from the cascade impactor to ensure atmospheric NO$_3^-$ concentrations were sufficient for isotope analysis (Sect. 2.2.2). However, this was not always possible as on occasion filters had to be removed early due to unusual ship manoeuvres that could have resulted in sample contamination by ship stack emissions if left unattended. Sampling duration ranged from 11 to 36 h TS2 in winter and 7 to 41 h in spring (Table S1 in the Supplement).

During each voyage, a field blank was collected by fitting the cascade impactor with a set of filters and loading the HV-AS in the same manner that atmospheric samples were deployed. The cascade impactor was then immediately removed without turning on the HV-AS pump. The field blanks were removed from the cascade impactor and stored in the same manner as the atmospheric samples. All chemical analysis performed on samples was performed on the field blanks to assess any possible contamination during filter deployment or laboratory procedures.

## 2.2 Sample analysis

Once back on land, filters were extracted using ultra-clean deionized water (DI; 18 MΩ) under a laminar-flow cabinet (Air Science). The extraction ratio was approximately 30 to 100 cm$^2$ of filter in 30 mL of DI. Extracts were immediately sonicated for 1 h and then stored at 4 °C for at least 12 h. Thereafter, extracts were filtered (0.2 µm) using an acid-washed syringe into clean 30 mL high-density polyethylene (HDPE) bottles and stored at $-20$ °C until analysis (Burger et al., 2022a).

### 2.2.1 NO$_3^-$ concentration analysis

[NO$_3^-$] was determined using a Thermo Scientific Dionex Aquion ion chromatography (IC) system equipped with an autosampler. The anion IC contained an AG22 RFIC $4 \times 50$ mm guard column and AG22 RFIC $4 \times 250$ mm analytical column. A six-point standard curve was run on each day of analysis (Dionex Seven Anion-II Standard), and an $R^2$ value $> 0.999$ was required for sample analysis to proceed. Final [NO$_3^-$ CE2 TS3 was corrected by subtracting the field blanks, which represented on average 32 % and 59 % of the [NO$_3^-$] in winter and spring, respectively. Where the field blank had a [NO$_3^-$] greater than that of the sample, the sample [NO$_3^-$] was assumed to be zero. Samples were measured for [NO$_3^-$] only once to preserve sample volume for isotopic analysis (Sect. 2.2.2), motivated by the small difference between repeated sample measurements from the summertime dataset (SD$_p = 0.3$ µmol L$^{-1}$). Only samples with concentrations greater than 1 µmol L$^{-1}$ where able to be measured for isotopic composition. Only samples with concentrations greater than 1 µmol L$^{-1}$ were able to be measured for isotopic composition. TS4

### 2.2.2 Isotopic analysis

The isotopic composition of atmospheric NO$_3^-$ ($\delta^{15}$N, $\delta^{18}$O, and $\Delta^{17}$O–NO$_3^-$) was measured using the denitrifier method (Sigman et al., 2001; Casciotti et al., 2002; Kaiser et al., 2007). In brief, a natural strain of denitrifying bacteria, *Pseudomonas aureofaciens*, that lack the terminal nitrous oxide (N$_2$O) reductase enzyme were used to convert aqueous NO$_3^-$ quantitatively to N$_2$O gas. The product N$_2$O was analysed by gas-chromatograph isotope ratio mass spectrometry (IRMS) (Thermo-Scientific Delta V Plus) for simultaneous isotopic determination of $^{15}$N / $^{14}$N and $^{18}$O / $^{16}$O (Sigman et al., 2001; Casciotti et al., 2002). The $^{15}$N / $^{14}$N of samples was corrected for the contribution of $^{17}$O to the peak at mass 45 using $\Delta^{17}$O determined for each sample, with values ranging from 21.7 ‰ to 44.4 ‰. International reference materials (Table S2) IAEA-N3 and USGS34 were used to normalize isotopic values to air ($\delta^{15}$N), and IAEA-N3, USGS34, and USGS35 were used to normalize to VSMOW ($\delta^{18}$O) scales. The pooled standard deviation of sample replicates

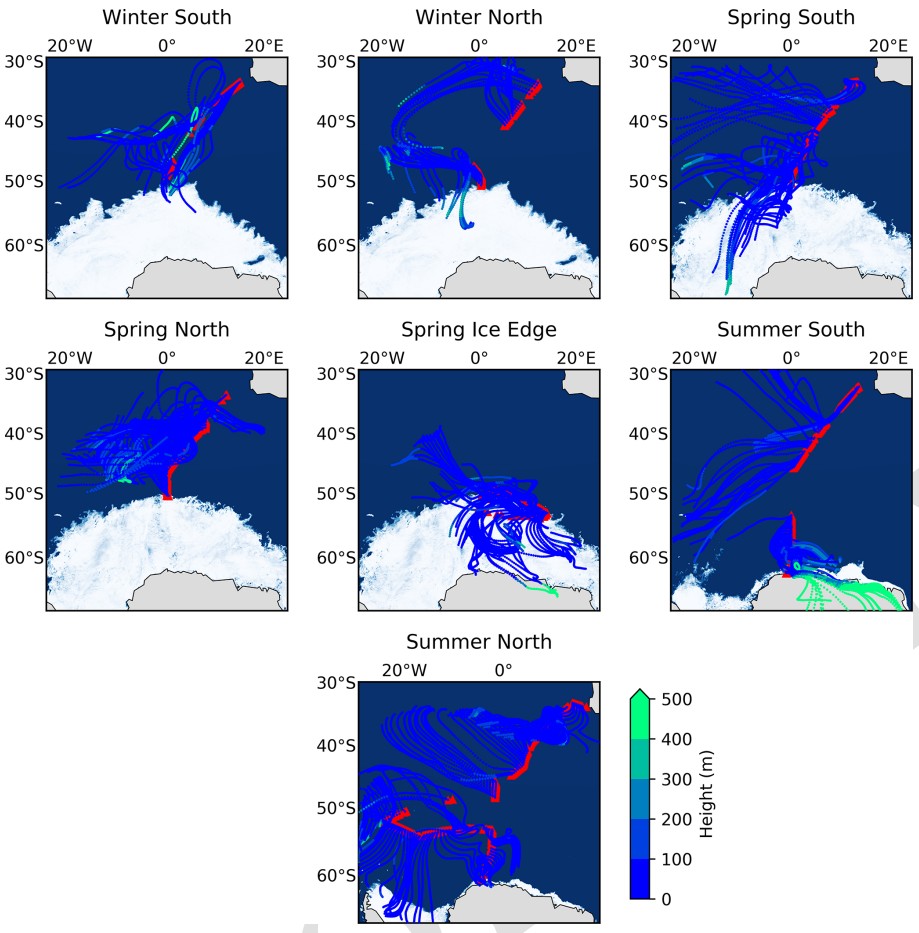

**Figure 1.** The 72 h air mass back trajectories (AMBTs) computed for each hour of every filter deployment made in winter (19 July to 12 August 2019) on both the southbound (**a**: "Winter South") and northbound (**b**: "Winter North") voyages; in spring (13 October to 19 November 2019) on the southbound voyage (**c**: "Spring South"), northbound voyage (**d**: "Spring North"), and ice edge transect (**e**: "Spring Ice Edge"); and in summer (7 to 21 December 2018 and 27 February to 15 March 2019) on the southbound (**f**: "Summer South") and northbound (**g**: "Summer North") voyages. Red triangles indicate the ship's cruise track during each filter deployment. The AMBTs are coloured by height (m) (blue-to-green colour bar).

and duplicates for $\delta^{15}$N was 0.19‰ ($n = 16$), and for $\delta^{18}$O it was 0.27‰ ($n = 16$). The pooled standard deviations of sample references IAEA-N3, USGS34, and USGS35 for $\delta^{15}$N and for $\delta^{18}$O are reported for each season in Table S3.

For winter and spring samples, $\Delta^{17}$O–NO$_3^-$ was characterized by using a separate 50 nmol aliquot to convert NO$_3^-$ to N$_2$O, thermally decomposing the N$_2$O to N$_2$ and O$_2$ in a gold furnace at 770 °C and analysing the isotopic composition of O$_2$ for determination of $^{18}$O / $^{16}$O and $^{17}$O / $^{16}$O (Kaiser et al., 2007; Fibiger et al., 2013). The product O$_2$ was referenced to USGS34 and USGS35, and a 50/50 mix of USGS34 and USGS35 was also quantified within runs serving as a quality control check. The pooled standard deviations for $\Delta^{17}$O were 0.84‰ ($n = 21$), 0.90‰ ($n = 21$), and 0.61‰ ($n = 18$) for USGS34, USGS35, and the 50/50 mix, respectively. The pooled standard deviation of sample

replicates and duplicates was 0.63‰ in winter and 0.31‰ in spring.

It is important to note that given the low [NO$_3^-$] of the field blanks ($< 1.5\,\mu$M), no isotopic analysis could be performed on the blank filters, and therefore the blank was not subtracted from the isotope results. However, we note that there was no relationship found between the blank percent contribution and $\delta^{15}$N–NO$_3^-$ or $\delta^{18}$O–NO$_3^-$ for spring and winter. This indicates that the measured signal is not driven by the blank contribution.

### 2.2.3 Seawater sampling and NO$_2^-$ concentration analysis

Seawater samples were collected in triplicate every 2 h from the ship's underway system (position at depth approximately 5 m) for the analysis of surface ocean nitrite concentrations ([NO$_2^-$]). Seawater samples for NO$_2^-$ determination were im-

mediately frozen at $-20\,°C$ and stored in dark conditions until analysis. $[NO_2^-]$ was analysed using the colorimetric method of Strickland and Parsons (1972) and Parsons et al. (1984) with a Thermo Scientific Genesys 30 visible spectrophotometer (detection limit of $0.05\,\mu mol\,L^{-1}$). The majority of seawater $[NO_2^-]$ analysis was conducted while at sea.

## 2.3   Air mass back trajectory analysis

Air mass back trajectories (AMBTs) were computed for each hour in which the HV-AS was operational for at least 45 min of that hour. Given that the ship was moving, a different date, time, and starting location were used to compute each AMBT. An altitude of 20 m was chosen to match the height of the HV-AS above sea level, and 72 h AMBTs were computed to account for the lifetime of $NO_3^-$ in the atmosphere. Model estimates of the atmospheric lifetime of $NO_3^-$ range from approximately 3 to 5 d (Lu et al., 2021). AMBTs become increasingly uncertain the further back in time they are used (Sinclair et al., 2013), particularly in the remote Southern Hemisphere. To minimize this uncertainty, the shortest possible AMBTs are generated while still accounting for the lifetime of $NO_3^-$ (i.e. 72 h). Daily 120 h AMBTs computed for the duration of each voyage were additionally computed (see Fig. S1 in the Supplement) to confirm that even when utilizing the maximum estimate for $NO_3^-$ atmospheric lifetime, no continental influence from southern Africa is expected. All AMBTs were computed with NOAA's Hybrid Single-Particle Lagrangian Integrated Trajectory (HYSPLIT) model (Stein et al., 2015; Rolph et al., 2017) using NCEP Global Data Assimilation System (GDAS) output, which can be accessed at https://www.ready.noaa.gov/index.php (last access: 3 May 2023) (NOAA Air Resources Laboratory, Silver Spring, Maryland).

## 3   Results and discussion

AMBTs indicate that no samples experienced any continental influence from South Africa (Fig. 1), such that no direct anthropogenic emission sources are considered. The 72 h AMBTs confirm that the Atlantic sector of the Southern Ocean was a dominant source region for most samples collected throughout all seasons. Air masses experienced very little interaction with sea ice in winter (Fig. 1a and b), while extensive interaction with sea ice was experienced by air masses sampled in spring, particularly at the high latitudes during the southbound leg (Fig. 1c) and ice edge transect (Fig. 1e). In summer, some high-latitude air masses traversed coastal Antarctica before being sampled, particularly on the southbound leg (Fig. 1f), while some interaction with sea ice was also experienced by high-latitude air masses on both legs (Fig. 1f and g). The potential for sea ice influence is supported by the relatively low height ($<100$ m) of AMBTs (Fig. 1). As a result, air masses originated from a mixture of source regions ranging from the open ocean to sea ice to

Antarctic continental ice. The remoteness of all the locations from which air masses originated motivates the investigation of natural $NO_x$ sources below.

## 3.1   Seasonal variation in atmospheric $NO_3^-$ concentrations

In winter, atmospheric $[NO_3^-]$ was very low across the Atlantic Southern Ocean, ranging from below TS5 detection to $22.3\,ng\,m^{-3}$ (Fig. 2a, blue diamonds). A single outlier exists with a relatively high $[NO_3^-]$ equivalent to $222.9\,ng\,m^{-3}$ in winter, although it is comparable to summertime $[NO_3^-]$ (Fig. 2a, orange circles). In winter only 6 of 12 filter sample deployments contained enough $NO_3^-$ for isotope analyses to proceed. Filter deployments containing no $NO_3^-$ or sample concentrations $<1\,\mu mol\,L^{-1}$ were excluded from Fig. 2a. In spring, atmospheric $[NO_3^-]$ ranged from 3.3 to $74.4\,ng\,m^{-3}$. Higher $[NO_3^-]$ was observed at the lower latitudes and at the higher latitudes, while lower $[NO_3^-]$ was observed in the mid-latitude Atlantic Southern Ocean (Fig. 2a, green squares). During summer atmospheric $[NO_3^-]$ was higher than winter and spring, ranging from 19.9 to $264.0\,ng\,m^{-3}$. In contrast to winter and spring, a distinct latitudinal trend was observed in summer whereby the $[NO_3^-]$ decreased with increasing latitude (Fig. 2a, orange circles) (Burger et al., 2022a).

The seasonal cycle in atmospheric $[NO_3^-]$ that we observe, i.e. lowest concentrations in winter, higher in spring, and highest in summer, is similar to previous observations for the region. Atmospheric $[NO_3^-]$ ranging from tens of nanogrammes per cubic metre to approximately $100\,ng\,m^{-3}$ have been observed for the Southern Ocean MBL during late spring (Morin et al., 2009; Shi et al., 2021), and observations from coastal Antarctic sites in the Atlantic sector showed elevated $[NO_3^-]$ ($\sim 20$ to $70\,ng\,m^{-3}$) in late spring and early summer (Wagenbach et al., 1998; Wolff et al., 2008). Seasonal studies at coastal and inland Antarctic sites observed the lowest $[NO_3^-]$ during winter (Wagenbach et al., 1998; Savarino et al., 2007; Wolff et al., 2008; Ishino et al., 2017; Walters et al., 2019).

The seasonality in atmospheric $[NO_3^-]$ is largely driven by the seasonality in sunlight availability. Maximum atmospheric $[NO_3^-]$ observed in late spring/early summer in coastal Antarctica was attributed to reactive N released from the post-depositional processing/recycling of snow $NO_3^-$ (Savarino et al., 2007). After $NO_3^-$ is deposited to the snowpack, it can be photochemically reduced to $NO_x$ and (re)emitted to the overlying atmosphere (Jones et al., 2000, 2001). During winter, extended periods of darkness lead to reduced photochemical activity above the snow, resulting in background-level $[NO_3^-]$ (Lee et al., 2014). Over the open ocean, increased UV radiation in spring and summer compared to winter may lead to greater $NO_3^-$ production from the photolytically derived oceanic $RONO_2$ source (Fisher et al., 2018). Ground-based studies in Antarctica demonstrate that

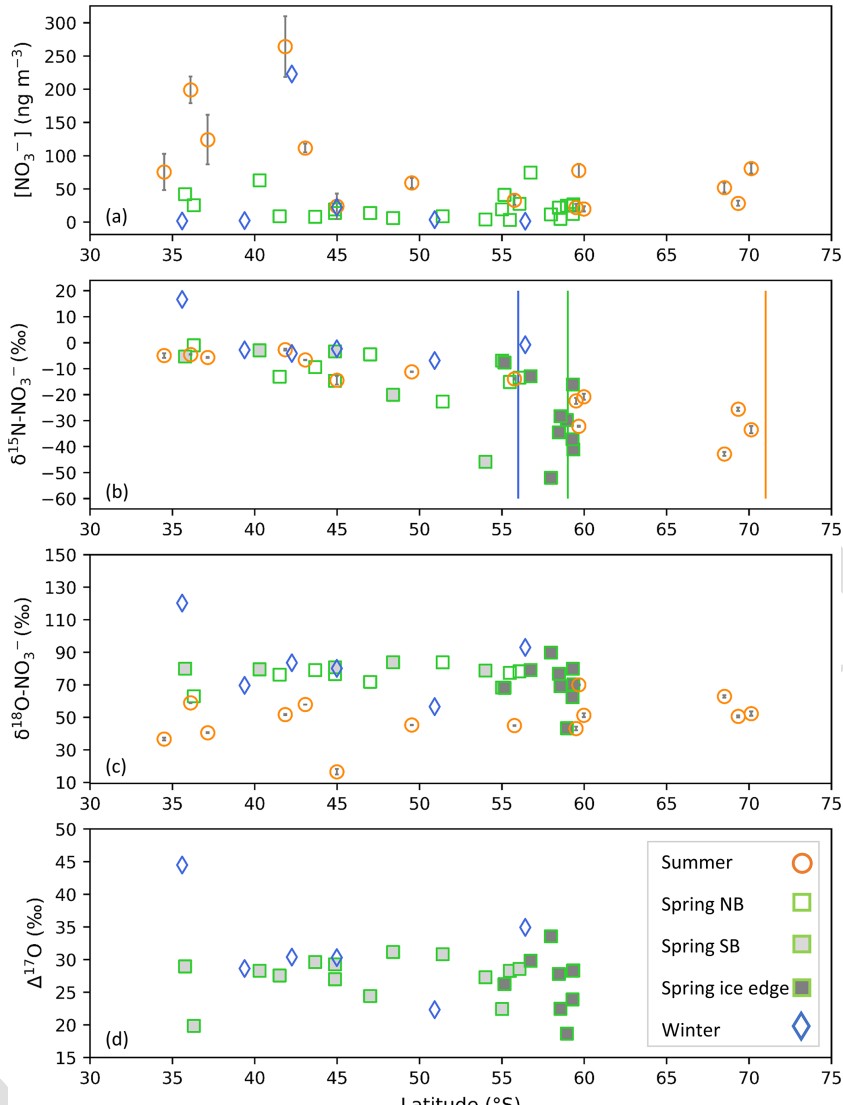

**Figure 2.** The average coarse-mode ($> 1\,\mu m$) atmospheric nitrate concentration [NO$_3^-$] (ng m$^{-3}$) **(a)**, weighted average $\delta^{15}$N of atmospheric nitrate ($\delta^{15}$N–NO$_3^-$ (‰ vs. N$_2$)) **(b)**, $\delta^{18}$O of atmospheric nitrate ($\delta^{18}$O–NO$_3^-$ (‰ vs. VSMOW)) **(c)**, and $\Delta^{17}$O of atmospheric nitrate ($\Delta^{17}$O–NO$_3^-$ (‰)) **(d)** as a function of latitude (° S). Winter, spring, and summer are denoted by blue diamonds, green squares, and orange circles, respectively. For the summer data, where error bars ($\pm 1$ SD) are not visible, the standard deviation is smaller than the size of the marker. Spring data are separated into northbound (NB), southbound (SB), and ice edge legs by clear, light-grey, and dark-grey fills, respectively for panels **(b)**–**(d)**. Vertical lines indicate the approximate location of the sea ice edge in summer (orange), winter (blue), and spring (green), identified visually using satellite-derived sea ice concentration obtained from passive microwave sensors of AMSR2 (Advanced Microwave Scanning Radiometer 2; Spreen et al., 2008).

UV radiation is highest in spring and early summer, when stratospheric O$_3$ concentrations are at a minimum, and the noon solar zenith angle is low (Aun et al., 2020; Lakkala et al., 2020). This, in addition to greater lightning NO$_x$ production during spring and summer at the lower southern latitudes ($< 40°$ S) (Nesbitt et al., 2000), likely explains why higher [NO$_3^-$] is observed in spring and summer as compared to winter.

## 3.2 Seasonal variation in NO$_x$ sources

While [NO$_3^-$] provides valuable information regarding the seasonal and spatial variation in the quantity of tropospheric NO$_3^-$ present, the N isotopic composition serves as a useful tool for identifying NO$_x$ sources that lead to aerosol NO$_3^-$ formation. Here, we present and interpret the mass-weighted, coarse-mode average $\delta^{15}$N–NO$_3^-$, computed for each filter deployment. In remote environments where O$_3$ concentrations largely exceed NO$_x$ concentrations, as is the case for

the remote Southern Ocean, NO$_x$ isotopic exchange occurs at a much slower rate than the Leighton cycle reactions. Therefore, little to no equilibrium isotope fractionation is expressed, and the $\delta^{15}$N of NO$_3^-$ is assumed to reflect the $\delta^{15}$N of the NO$_x$ source (Walters et al., 2016).

### 3.2.1 Evidence for stratospheric NO$_3^-$

There was one unusually high $\delta^{15}$N–NO$_3^-$ value equivalent to 16.6‰ for the first filter deployment of the southbound leg in winter (Fig. 2b). Despite an elevated $\delta^{15}$N signature, its [NO$_3^-$] (1.7 ng m$^{-3}$) was consistent with that of most wintertime samples. The $\delta^{15}$N–NO$_3^-$ of this wintertime sample is similar to the $\delta^{15}$N of stratospherically sourced NO$_3^-$, estimated to be $19 \pm 3$‰ (Savarino et al., 2007). Stratospheric input is additionally supported by the air mass history of this sample, which indicates that air originated from as far south as the sea ice edge for the duration of the sample deployment (Fig. 1a). Near the sea ice edge, some AMBTs originate from greater heights (300 to 400 m) TS6 and descend towards the sampling location (< 100 m) (Fig. S2). Coastal Antarctic studies suggest that the deposition of polar stratospheric clouds (PSCs) during winter results in stratospheric NO$_3^-$ inputs to the Antarctic troposphere (Wagenbach et al., 1998; Savarino et al., 2007). Winter, when this sample was collected, is the only time of year when Antarctic temperatures are expected to be cold enough (< 195 K) for PSC formation (von Savigny et al., 2005; Wang et al., 2008).

Furthermore, this sample is unique in that it has a relatively high $\delta^{18}$O–NO$_3^-$ and $\Delta^{17}$O–NO$_3^-$, 120.2‰ and 44.5‰, respectively. Tropospheric oxidation typically produces $\Delta^{17}$O–NO$_3^-$ values ranging from 17.3‰ to 42.7‰ (Morin et al., 2011; Ishino et al., 2017; Walters et al., 2019). Stratospherically sourced $\Delta^{17}$O–NO$_3^-$ is elevated in comparison to tropospheric $\Delta^{17}$O–NO$_3^-$ because stratospheric O$_3$ has a greater isotope anomaly than tropospheric O$_3$, and/or dominance of the N$_2$O$_5$ and ClONO$_2$ pathways allows for greater transfer of the anomaly to NO$_3^-$ via O$_3$ (Savarino et al., 2007; McCabe et al., 2007). High $\Delta^{17}$O–NO$_3^-$ values (> ∼ 40‰) observed in winter are often attributed to contributions by stratospheric denitrification (Savarino et al., 2007; McCabe et al., 2007; Frey et al., 2009; Walters et al., 2019). The combination of elevated $\Delta^{17}$O–NO$_3^-$, $\Delta^{18}$O–NO$_3^-$, and $\delta^{15}$N–NO$_3^-$ is consistent with a stratospheric NO$_3^-$ source for this sample. Given the evidence that this sample likely does not reflect tropospheric oxidation chemistry, it is left out of the analysis below.

### 3.2.2 Transported NO$_x$

Previous modelling studies suggest that tropospheric transport of NO$_x$ emitted in the mid-latitudes to low latitudes (i.e. soil, lightning, thermal decomposition of PAN, and fossil fuel combustion), contributes to the Antarctic NO$_3^-$ budget in winter (Lee et al., 2014). PAN decomposition has previously been suggested as a NO$_x$ source to coastal Antarctica during winter and early spring (Savarino et al., 2007; Jones et al., 2011). However, transported NO$_x$ results in minimal NO$_3^-$, regarded as background-level concentrations (<∼ 10 ng m$^{-3}$; Savoie et al., 1993; Lee et al., 2014), consistent with most of our winter observations (Fig. 2a, blue diamonds). During winter, $\delta^{15}$N–NO$_3^-$ was relatively invariant across the Atlantic Southern Ocean (Fig. 2b, blue diamonds), with an average of $-3.4 \pm 2.1$‰ ($n = 5$). This is consistent with a lack of snowpack NO$_x$ emissions at the high latitudes during July/August due to weak or absent solar radiation (Shi et al., 2022), as well as minimal influence from oceanic RONO$_2$. Furthermore, air mass back trajectory analyses indicate that sea ice had a very minor influence on the winter samples (Fig. 1a and b).

Albeit outside of the winter months, previous studies report an average $\delta^{15}$N–NO$_3^-$ for the low-latitude Atlantic Ocean (between 45° S and 45° N) on the order of $-3$‰ to $-4$‰ (Baker et al., 2007; Morin et al., 2009), attributed to a combination of natural NO$_x$ sources including lightning, biomass burning, and soil emissions (Morin et al., 2009). This is also similar to the spring observations, where higher values of $\delta^{15}$N–NO$_3^-$ were observed at the lower latitudes ($-3.2 \pm 1.8$‰, $n = 3$), i.e. equatorward of 40° S. As such, the winter samples and low-latitude spring samples could be representative of a combination of natural NO$_x$ sources emitted further north and transported to the mid- to low-latitude Atlantic Ocean.

Not all winter samples isotopically indicative of the transported background NO$_x$ source had low [NO$_3^-$]. One unusually high [NO$_3^-$] value (222.9 ng m$^{-3}$) was observed at the lower latitudes (Fig. 2a, blue diamonds). Due to the similarity in isotopic composition among winter samples, we can assume that despite a higher [NO$_3^-$], this sample also originated from a combination of natural NO$_x$ sources transported from the lower latitudes. Furthermore a [NO$_3^-$] on the order of 200 ng m$^{-3}$ is consistent with summertime [NO$_3^-$] observations (Fig. 2a, orange circles), when natural NO$_x$ sources dominated (see Sect. 3.2.3). Our results thus confirm that, like in summer, natural NO$_x$ sources can at times lead to relatively high [NO$_3^-$], even in winter, when background conditions are typically experienced.

In addition, the winter dataset presented here clearly highlights the utility of the isotopes in distinguishing NO$_x$ sources. The initial winter sample had a low concentration indicative of the background conditions; however, the triple-stable-isotopic composition of the sample confirms that it originated from the stratosphere (see Sect. 3.2.1). In contrast, the anomalously high [NO$_3^-$] sample observed in winter was not consistent with minimal background NO$_x$ emissions; however its $\delta^{15}$N confirmed that this was in fact the most likely source.

### 3.2.3 Snowpack photolysis and oceanic NO$_x$ sources

Springtime $\delta^{15}$N–NO$_3^-$ ranged from $-52.0\permil$ to $-1.1\permil$, and samples with the lowest $\delta^{15}$N–NO$_3^-$ were observed at the high latitudes (Fig. 2b, green squares). The range in $\delta^{15}$N–NO$_3^-$ observed for spring is consistent with late-spring/early-summer (November to December) observations from the Indian Ocean sector of the Southern Ocean (Shi et al., 2021) and summer (December and March) observations from the Atlantic sector (Burger et al., 2022a). Springtime $\delta^{15}$N–NO$_3^-$ is also consistent with long-term records of $\delta^{15}$N–NO$_3^-$ measured in coastal Antarctica (Wagenbach et al., 1998) and on the East Antarctic Plateau (Winton et al., 2020) for the same season. Given the similarity in $\delta^{15}$N–NO$_3^-$ between spring and summer, we expect the dominant NO$_x$ sources to be the same.

During spring, air mass back trajectories indicate substantial sea ice influence at the high latitudes during the southbound leg and during the ice edge transect (Fig. 3a and c). There is a large isotope effect associated with snow NO$_3^-$ photolysis during summer in the Antarctic (Berhanu et al., 2014, 2015; Frey et al., 2009; Erbland et al., 2013), resulting in the emission of low $\delta^{15}$N–NO$_x$ ($\sim -48\permil$) to the overlying atmosphere (Savarino et al., 2007; Morin et al., 2009; Shi et al., 2018; Walters et al., 2019). The low $\delta^{15}$N–NO$_3^-$ samples from the high latitudes (minimum $-52.0\permil$) are clearly influenced by sea ice (Figs. 2b and 3a, c), but the air masses do not cross the Antarctic continent. This suggests that the low $\delta^{15}$N–NO$_x$ likely comes from snow nitrate photolysis from the snow on sea ice before a net loss of NO$_3^-$ from the snowpack leads to any large $^{15}$N enrichment in the snow and subsequently the atmosphere (Shi et al., 2018). We conclude that NO$_x$ as a result of photolysis of snow nitrate on sea ice can explain the relatively low $\delta^{15}$N–NO$_3^-$ observed in samples collected at the high latitudes on the spring southbound leg and during the ice edge transect (Fig. 2b, filled grey squares).

Higher $\delta^{15}$N–NO$_3^-$ values ($-22.7\permil$ to $-1.0\permil$) were observed during spring for the northbound leg (Fig. 2b, open squares; Fig. 3b). The $\delta^{15}$N of atmospheric NO$_3^-$ that originates from snowpack emissions depends on the $\delta^{15}$N of the local snowpack NO$_x$ source. $^{15}$N enrichment in the snow due to preferential loss of $^{14}$N during photolysis can eventually lead to increased $\delta^{15}$N–NO$_x$ and ultimately higher values of atmospheric $\delta^{15}$N–NO$_3^-$ (Shi et al., 2018). However, the air mass histories of the samples indicate no contact with surrounding sea ice (i.e. the northbound leg; Fig. 2b, open squares; Fig. 3b), suggesting that any influence from snowpack NO$_x$ emissions was limited. These samples originated from over the mid-latitude region of the Southern Ocean, where detectable sea surface nitrite was present (Fig. 3b). The NO which originates from nitrite in seawater is thought to limit sea surface RONO$_2$ production. As a result, elevated nitrite concentrations are required for RONO$_2$ production

to occur in seawater (Dahl and Saltzman 2008; Dahl et al., 2012). Oceanic RONO$_2$ has been long proposed as an important primary NO$_3^-$ source to the Antarctic (Jones et al., 1991). Recent studies have used modelling and isotopic approaches to investigate the relative importance of oceanic RONO$_2$ compared to other sources of NO$_3^-$ in the Southern Ocean MBL, particularly in summer (Fisher et al., 2018; Burger et al., 2022a). However, limited co-occurring ocean atmosphere measurements are available to constrain the seasonality of the RONO$_2$ source. While $\delta^{15}$N–RONO$_2$ has yet to be directly quantified, it was recently estimated to have an average $\delta^{15}$N signature of $\sim -22\permil$ TS7 in the summertime Southern Ocean (Burger et al.,2022a) and $-27.8\permil$ in the eastern equatorial Pacific (Joyce et al., 2022). Consistent with this relatively isotopically light oceanic RONO$_2$ source are observations of relatively low aerosol $\delta^{15}$N–NO$_3^-$ from the mid-latitude Southern Ocean (Burger et al., 2022a) and eastern equatorial Pacific (Kamezaki et al., 2019; Joyce et al., 2022), on the order of $-15\permil$ to $-7\permil$.

Trends in $\delta^{15}$N–NO$_3^-$ by air mass origin were most evident in the ice edge transect, during which lower (higher) $\delta^{15}$N–NO$_3^-$ values were observed for samples with greater sea ice (oceanic) influence (Fig. 3c). The photolysis imprint on the NO$_3^-$ stable isotope signal in the marine boundary layer above the sea ice is clearly observed and speaks to the importance of snow-covered sea ice as a NO$_x$ source in the region during spring as well as summer. The increased importance of oceanic RONO$_2$ emissions as the air mass origin migrates from sea-ice-covered to open-ocean zones is also evidenced by the decrease TS8 in $\delta^{15}$N–NO$_3^-$ observed for air mass originating predominantly from over the ocean (Fig. 3c).

Isotopically, there is little evidence of RONO$_2$ emissions contributing to aerosol NO$_3^-$ in the winter samples. Reduced levels of UV radiation and minimal daylight hours (Fig. S3) in winter likely hinder the contribution of the oceanic NO$_x$ source to NO$_3^-$ loading compared to spring/summer, despite detectable sea surface nitrite concentrations in winter (Fig. S4). Additionally, photolysis in spring/summer serves to produce OH, which is the primary oxidant for converting RONO$_2$-derived NO$_x$ to NO$_3^-$ in the MBL (Fisher et al., 2018).

Some studies suggest that the photolysis of particulate NO$_3^-$ (p-NO$_3^-$) associated with sea-salt aerosols in the MBL can serve as an important NO$_x$ source (Zhou et al., 2003; Ye et al., 2016; Reed et al., 2017). However, the importance of this NO$_x$ formation pathway remains unclear, with large variability in reported rates between studies (Ye et al., 2016; Reed et al., 2017; Kasibhatla et al., 2018; Romer et al., 2018). To our knowledge, there are no observations of p-NO$_3^-$ photolysis from the Southern Ocean MBL, and the implications of this process on the isotopic composition of NO$_3^-$ in the MBL have yet to be assessed. We know that NO$_3^-$ photolysis in snow is associated with a large fractionation, leading to

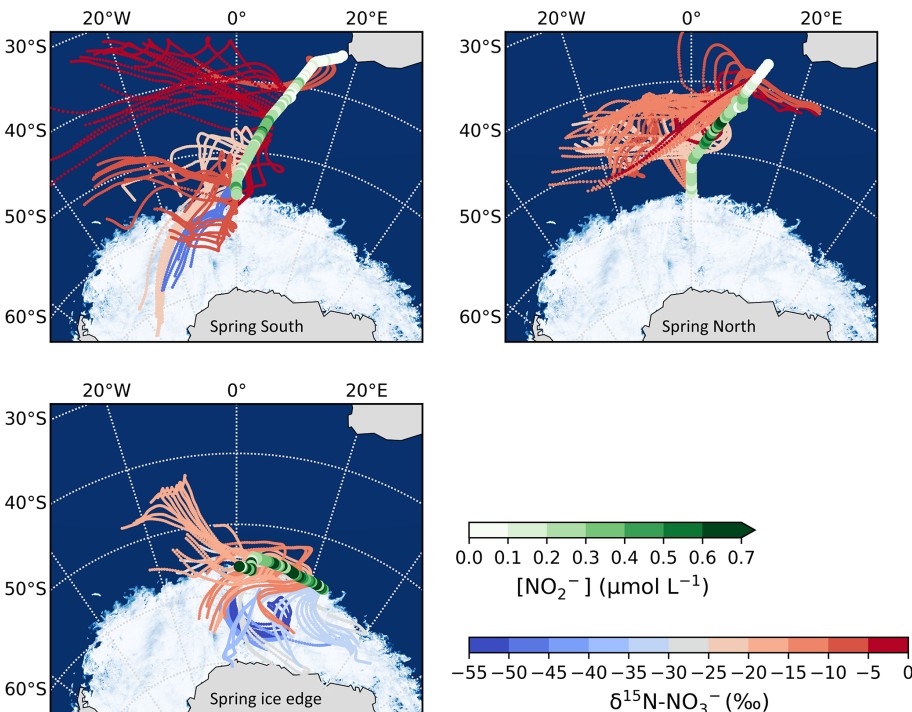

**Figure 3.** The 72 h AMBTs computed for each hour of the spring cruise during **(a)** the southbound leg ("Spring South"), **(b)** the northbound leg ("Spring North"), and **(c)** the ice edge transect ("Spring ice edge") when the HV-AS was running for more than 45 min of the hour. AMBTs are colour-coded by the weighted average $\delta^{15}$N–NO$_3^-$, represented by the blue-to-red colour bar. Overlaid are the surface ocean nitrite concentrations ([NO$_2^-$]), represented by the green colour bar. The white area represents the location of the sea ice determined using satellite-derived sea ice concentration data obtained from the passive microwave sensors of AMSR2 (Advanced Microwave Scanning Radiometer 2; Spreen et al., 2008).

the emission of isotopically light NO$_x$, while the remaining NO$_3^-$ pool becomes enriched in $^{15}$N (e.g. Frey et al., 2009; Berhanu et al., 2014, 2015; Shi et al., 2018). Thus, if the p-NO$_3^-$ we measured were affected by photolysis we would have expected to observe much higher or even positive values of $\delta^{15}$N–NO$_3^-$ during spring and summer. Another scenario is that the p-NO$_3^-$ we measured resulted from the oxidation of NO$_x$ released by prior p-NO$_3^-$ photolysis. In this case, we would have expected to observe much lower $\delta^{15}$N–NO$_3^-$ values over the open ocean, on par with those observed over the ice. Since neither of the above scenarios matches the observations, the potential influence of aerosol NO$_3^-$ photolysis as a significant NO$_x$ source to the region during our study is unlikely.

Additionally, a strong anti-correlation ($r = -0.86$) is observed between $\delta^{15}$N–NO$_3^-$ and $\Delta^{17}$O–NO$_3^-$ for samples collected in spring which experience a sea ice influence greater than 75 % (Fig. 4), determined based on air mass history. A similar relationship was observed at Dome C during summer (Erbland et al., 2013; Savarino et al., 2016). Previous studies found that the production of enhanced $\Delta^{17}$O–NO$_3^-$ in polar regions is linked to the intensity of NO$_x$ emissions from the snowpack (Morin et al., 2012; Savarino et al., 2016). The correlation between $\Delta^{17}$O–NO$_3^-$ and $\delta^{15}$N–NO$_3^-$ could arise

from an increased contribution of HONO photolysis to total OH production, which is co-emitted with NO$_x$ from the snowpack (Grannas et al., 2007; Legrand et al., 2014) and induces a greater $^{17}$O excess in OH compared to the OH production pathway: O($^1$D) + H$_2$O (Savarino et al., 2016). It could also arise from the coupling of snowpack emissions with reactive halogen chemistry as suggested by Morin et al. (2012). The relationship between $\Delta^{17}$O and $\delta^{15}$N presented here for the spring samples with air mass histories that indicate extensive influence from snow-covered sea ice suggests that snowpack emissions may lead to enhanced $\Delta^{17}$O transfer to NO$_3^-$.

## 3.3 Seasonal variation in oxidation

As mentioned in Sect. 1, the oxidation of NO and NO$_2$ can be determined using the oxygen isotopic composition of aerosol NO$_3^-$. Here, we present and interpret the mass-weighted, coarse-mode average $\delta^{18}$O–NO$_3^-$ and $\Delta^{17}$O–NO$_3^-$, computed for each filter deployment.

During NO and NO$_2$ oxidation, the oxygen atoms of the responsible oxidants are incorporated into the NO$_3^-$ product. The transferrable terminal oxygen atom of O$_3$ possesses an elevated $\Delta^{17}$O–NO$_3^-$(O$_{3\text{term}}$) and $\delta^{18}$O(O$_{3\text{term}}$) (39.3 ± 2‰

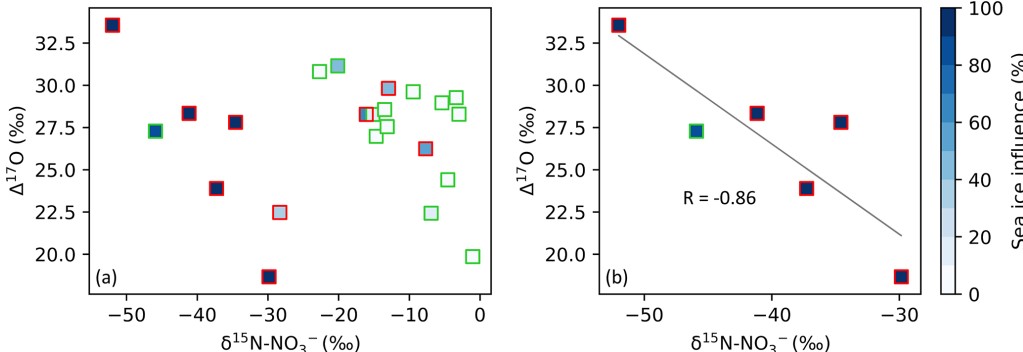

**Figure 4.** The relationship between $\Delta^{17}$O–NO$_3^-$ and $\delta^{15}$N–NO$_3^-$ in spring (square symbols). In both panels, samples collected along the ice edge are denoted by the red edge colour, with all other samples collected on the north- and southbound legs of the voyage denoted by the green edge colour. The colour bar (blues) indicates the percentage of sea influence experienced by each filter sample as determined using AMBTs. In panel **(a)**, all spring samples are included. In panel **(b)**, only samples that experienced a sea ice influence $> 75\%$ are included. A straight line (grey) is fitted to the data in panel **(b)**. Note the difference in $x$-axis scale between panels.

and $126.3 \pm 11.9\%_0$, respectively) (Vicars and Savarino, 2014) compared to other oxidants (e.g. OH and peroxy radicals, RO$_2$ / HO$_2$) which possess a $\Delta^{17}$O–NO$_3^- \approx 0\%_0$ (Michalski et al., 2012). The $\delta^{18}$O of OH is negative, while the $\delta^{18}$O of RO$_2$ / HO$_2$ stems from that of atmospheric O$_2$, which is also low (23.9 ‰; Barkan and Luz 2005). These differences allow us to qualitatively assess NO and NO$_2$ oxidation chemistry involving contributions by various oxidants. Similar to previous work conducted in the Southern Ocean MBL and in Antarctica (Walters et al., 2019; Shi et al., 2021), we make the assumption that oxidant $\delta^{18}$O values are known and directly represented in the NO$_3^-$.

The relatively low $\delta^{18}$O–NO$_3^-$ values observed in summer ($< 70\%_0$; Fig. 2c) are consistent with NO$_2$ oxidation via OH (Burger et al., 2022a). During summer, unusually low $\delta^{18}$O–NO$_3^-$ values were also observed, equating to less than the minimum expected for the OH oxidation pathway ($< \sim 46\%_0$; Burger et al., 2022a). This was attributed to an increased contribution by HO$_2$ / RO$_2$ during NO oxidation to NO$_2$ (as opposed to O$_3$), which would decrease the $\delta^{18}$O of the product NO$_3^-$. Increased abundance of RO$_2$ in the MBL was attributed to RONO$_2$ photolysis, hypothesized to occur over the mid-latitude Southern Ocean (Fisher et al., 2018; Burger et al., 2022a), and/or the presence of sea ice, which can lead to enhanced peroxy radical production (Brough et al., 2019).

Interestingly, despite NO$_x$ sources being the same in spring and summer (Sect. 3.2), the $\delta^{18}$O–NO$_3^-$ data suggest that the NO$_3^-$ formation pathways differ (Fig. 2c). Higher average $\delta^{18}$O–NO$_3^-$ values were observed in spring compared to summer (Fig. 2c). Higher $\delta^{18}$O–NO$_3^-$ values in spring compared to summer may originate from NO$_x$ oxidation by XO. In the Antarctic boundary layer, enhanced levels of BrO occur in spring over sea-ice-covered areas (Theys et al., 2011). The production of inorganic bromine has been pro-

posed to be related to frost flowers on thin sea ice (Kaleschke et al., 2004) and blowing of saline snow on sea ice (Yang et al., 2010). Significant interaction with sea ice cover was experienced in spring, particularly at the ice edge transect, which could have promoted NO$_3^-$ formation via the BrO pathway, resulting in increased values of $\delta^{18}$O–NO$_3^-$.

The oxygen isotopic composition of NO$_3^-$ in winter and spring was comparable as indicated by both $\delta^{18}$O (Fig. 2c) and $\Delta^{17}$O (Fig. 2d). The $\delta^{18}$O–NO$_3^-$ ranged from 56.5 ‰ to 92.9 ‰ in winter (Fig. 2c, blue diamonds; excluding the initial sample of stratospheric origin) and from 62.3 ‰ [TS9] to 89.8 ‰ in spring (Fig. 2c, green squares). The $\Delta^{17}$O–NO$_3^-$ ranged from 22.3 ‰ to 35 ‰ in winter (Fig. 2d, blue diamonds; excluding the initial sample of stratospheric origin) and from 18.7 ‰ to 33.6 ‰ in spring (Fig. 2d, green squares). Interestingly, there is more variability in the $\delta^{18}$O and $\Delta^{17}$O for the ice edge transect (Fig. 2c and d, dark shaded squares) than the north- and southbound transects. The overlap in $\delta^{18}$O and $\Delta^{17}$O in winter and spring suggests that similar pathways lead to NO$_3^-$ formation in both seasons, i.e. oxidation pathways that result in an increased influence of O$_3$ during oxidation (i.e. N$_2$O$_5$, DMS, XO).

A significant linear relationship was observed between $\delta^{18}$O–NO$_3^-$ and $\Delta^{17}$O–NO$_3^-$ in both winter and spring (Fig. S5). This suggests isotopic mixing between two major oxidants (e.g. Fibiger et al., 2013; Shi et al., 2021). As such, the highest end-member is representative of tropospheric O$_3$ and/or XO with a $\delta^{18}$O of $\sim 114\%_0$ to 138 ‰ and a $\Delta^{17}$O of $\sim 39\%_0$. There are multiple options for the second oxidant with a $\Delta^{17}$O $= 0\%_0$, e.g. water vapour (H$_2$O$_{(v)}$), OH, and O$_2$. Here, we use the $\delta^{18}$O–H$_2$O$_{(v)}$ from the average of observations along a similar cruise transect from the Indian sector of the Southern Ocean (Dar et al., 2020). The average $\delta^{18}$O–H$_2$O$_{(v)}$ determined between $\sim 33$ and $\sim 60°$ S ($-13.9 \pm 1.4\%_0$) was used for the winter samples given that AMBTs indicate that most air masses originated within this

**Table 1.** A summary of the oxygen isotope ratios ($\delta^{18}$O and $\Delta^{17}$O) for the end-member oxidants and/or oxidant sources (O$_3$, OH, HO$_2$ / RO$_2$, and H$_2$O) utilized in Sect. 3.3.

| Oxidant/source | $\delta^{18}$O (‰) | References | $\Delta^{17}$O (‰) | References |
|---|---|---|---|---|
| Terminal O$_3$ | $126.3 \pm 11.9$ | Vicars and Savarino (2014) | $39.3 \pm 2$ | Vicars and Savarino (2014) |
| OH | $-52.7 \pm 2.8^*$ | Walters and Michalski (2015) | $\sim 0$ | Michalski et al. (2012) |
| HO$_2$ / RO$_2$ | $23.88 \pm 0.03$ | Barkan and Luz (2005) | $\sim 0$ | Michalski et al. (2012) |
| H$_2$O | $-13.9 \pm 1.4$ | Dar et al. (2020) | $\sim 0$ | Michalski et al. (2012) |

* The average $\delta^{18}$O–OH was calculated from the equilibrium fractionation between OH and H$_2$O$_{(v)}$ (Walters and Michalski, 2015) using the observed atmospheric temperature range for winter and spring and the average $\delta^{18}$O–H$_2$O (Dar et al., 2020).

latitudinal band, where there is minimal variation in $\delta^{18}$O–H$_2$O$_{(v)}$ (Dar et al., 2020). In spring, the zone of air mass origin for our samples extends further south to $\sim 70°$ S. As shown by Dar et al. (2020), $\delta^{18}$O–H$_2$O$_{(v)}$ declines significantly between $\sim 60$ and $\sim 70°$ S. To account for this lowering in $\delta^{18}$O–H$_2$O$_{(v)}$, which could influence higher-latitude samples, an additional H$_2$O$_{(v)}$ end-member equivalent to the minimum observed by Dar at al. (2020) ($-27.5$‰) was included for spring. The $\delta^{18}$O–OH was calculated from the equilibrium fractionation between OH and H$_2$O$_{(v)}$ (Walters and Michalski, 2015) using the observed atmospheric temperature range for winter and spring. The $\delta^{18}$O–OH determined for winter ranges from $-56.2$‰ to $-49.5$‰ (average $= -52.8$‰), and for spring it ranges from $-54.5$‰ to $-50.5$‰ (average $= -52.5$‰). Therefore, a value of $-53$‰ was used for both seasons. The atmospheric $\delta^{18}$O–O$_2$ is well constrained at 23.9‰ (Barkan and Luz, 2005). The $\delta^{18}$O and $\Delta^{17}$O values assumed for all oxidants or oxygen sources outlined above are summarized in Table 1. Mixing lines for the three oxidant pairs (OH and O$_3$, H$_2$O$_{(v)}$ and O$_3$, and O$_2$ and O$_3$) are indicated by the grey, orange, and red lines, respectively, in Fig. 5.

To determine the lower end-member in each season, i.e. the second major oxidant in addition to ozone and/or XO, a straight line was fitted to the data in $\delta^{18}$O–$\Delta^{17}$O space, and the $x$ intercept at a $\Delta^{17}$O $= 0$‰ was determined. The $x$ intercept in winter is $\sim -16$‰. During winter, the linear relationship observed (Fig. 5a) is similar to what has been seen in the Indian Ocean MBL and in coastal East Antarctica, where the $x$ intercept was $-11 \pm 8$‰ (Shi et al., 2021) and $-15 \pm 6$‰ (Shi et al., 2022), respectively. The oxygen isotopic composition of the lower end-member in our winter data is most similar to that of H$_2$O$_{(v)}$. This is consistent with the average $\delta^{18}$O–H$_2$O$_{(v)}$ ($= -13.9 \pm 1.4$‰) observed between approximately 33 and 60° S (Dar et al., 2020). Therefore, a mixing line between H$_2$O$_{(v)}$ and O$_3$ is the best fit to the winter observations (Fig. 5a, solid orange line). If we exclude an equilibrium isotope fractionation between OH and H$_2$O$_{(v)}$ (Michalski et al., 2012) such that $\delta^{18}$O–OH is similar to the $\delta^{18}$O of H$_2$O$_{(v)}$, then the lower end-member likely results from the OH oxidation pathway.

By contrast, observations made in spring are best represented by mixing between three major oxidants: H$_2$O$_{(v)}$, O$_3$, and O$_2$. The $x$ intercept in spring is $\sim -4$‰, making it more difficult to identify one low $\delta^{18}$O end-member. The oxidant source with the closest oxygen isotope composition is again H$_2$O$_{(v)}$, indicating the prevalence of the OH pathway (when $\delta^{18}$O–OH $\sim \delta^{18}$O–H$_2$O$_{(v)}$); however the $x$ intercept is greater in spring compared to winter, suggesting that the lower end-member has a higher $\delta^{18}$O. H$_2$O$_{(v)}$ data from the region suggest that we would not expect to see a $\delta^{18}$O $> -10$‰; therefore an increase in H$_2$O$_{(v)}$ $\delta^{18}$O from winter to spring can be ruled out. A more likely explanation is that the springtime lower end-member consists of some combination of H$_2$O$_{(v)}$ and an additional higher $\delta^{18}$O oxidant that is less abundant in winter. The higher $\delta^{18}$O oxidant is likely atmospheric O$_2$ ($\delta^{18}$O $= 23.9$‰, $\Delta^{17}$O $= 0$‰ vs. VSMOW; Barkan and Luz, 2005). This is consistent with the spread in the springtime observations, which are bound by the decreased mixing line of H$_2$O$_{(v)}$ and O$_3$ TS14 to accommodate the influence of lower $\delta^{18}$O–H$_2$O$_{(v)}$ at the high latitudes (Fig. 5b, orange line) and the CE3 TS15 mixing line of atmospheric O$_2$ and O$_3$ (Fig. 5b, red line).

The influence of atmospheric O$_2$ during spring likely results from the increased role of RO$_2$ (and/or HO$_2$) in NO$_x$ cycling. This may be linked to increased RO$_2$ production over the mid-latitude Southern Ocean, derived from RONO$_2$ photolysis in the MBL (Burger et al., 2022a). There is also evidence that sea ice can lead to enhanced peroxy radical production (Brough et al., 2019), resulting in the potential for increased RO$_2$ and HO$_2$ concentrations to be observed in air masses that traverse the sea ice zone before being sampled. $\delta^{18}$O–NO$_3^-$ is greater in winter and spring compared to summer (Fig. 2c), highlighting the increased control of O$_3$ on the oxygen isotopic composition of NO$_3^-$ in winter and spring. Consistent with increased O$_3$ influence are seasonally resolved observations of O$_3$ concentration ([O$_3$]) in coastal Antarctica (Ishino et al., 2017; Shi et al., 2022) and Cape Grim, Tasmania (Derwent et al., 2016), the latter being more representative of the MBL. In all cases, maximum [O$_3$] is observed in winter, and minimum [O$_3$] is observed throughout summer. In spring, [O$_3$] concentrations are noticeably reduced compared to the winter, but slightly elevated com-

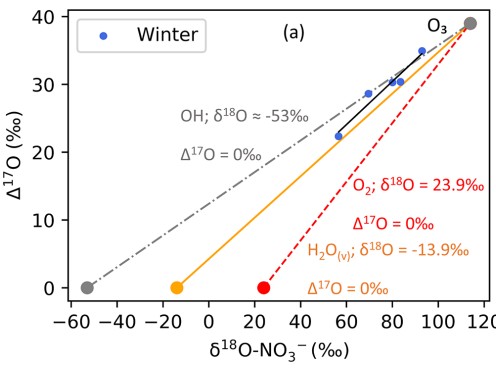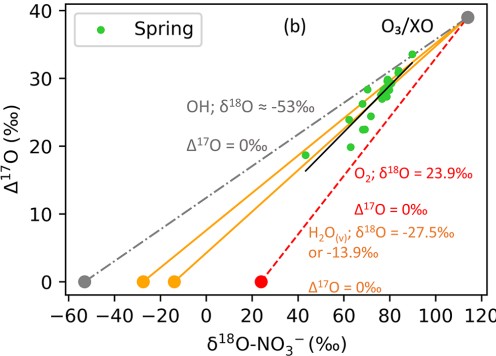

**Figure 5.** Winter and spring $\delta^{18}$O–NO$_3^-$ vs. $\Delta^{17}$O–NO$_3^-$ are plotted in panels **(a)** and **(b)**, respectively. A straight line (black) is fitted to the data in each panel. In both panels the grey line represents the mixing line of OH and O$_3$ TS10, the orange line represents the mixing line of H$_2$O$_{(v)}$ and O$_3$ TS11, and the red line represents the mixing line of O$_2$ and O$_3$ TS12. In panel **(b)**, an additional mixing line of H$_2$O$_{(v)}$ and O$_3$ TS13 is included (also in orange) to account for potentially lower values of $\delta^{18}$O–H$_2$O$_{(v)}$ ($\sim -27.5\,‰$) at 60 to 70° S.

pared to summer. Higher $\delta^{18}$O–NO$_3^-$ values in spring may also originate from NO$_x$ oxidation by XO, for example BrO, as discussed above.

## 4 Conclusions

Seasonally resolved observations of atmospheric NO$_3^-$ across the Atlantic Southern Ocean MBL suggest that natural NO$_x$ sources dominate throughout the year. Similar NO$_3^-$ sources are available to the MBL in both spring and summer, highlighting the importance of oceanic RONO$_2$ emissions in seasons other than the more frequently sampled summer months in the Southern Ocean. Although further research is required to improve our mechanistic and isotopic understanding of oceanic RONO$_2$ formation, fluxes, and conversion to aerosol NO$_3^-$, this work contributes to our growing understanding of how the surface ocean influences the atmospheric reactive N cycle and oxidation chemistry of the MBL (Altieri et al., 2021; Burger et al., 2022a; Joyce et al., 2022).

Furthermore, the large-sea-ice-extent characteristic of spring highlights the importance of snow-covered sea ice as a NO$_x$ source, in addition to the well-documented summer source from snow-covered continental ice (Jones et al., 2001; Walters et al., 2019; Winton et al., 2020). Currently no measurements of $\delta^{15}$N–NO$_3^-$ from snowpack on sea ice exist for Antarctica, which is an important measurement gap that should be addressed in future studies. The presence of sea ice may also play a role in the formation of peroxy radicals through its influence on chlorine chemistry when sunlight is available (Brough et al., 2019). Peroxy radicals (RO$_2$), H$_2$O$_{(v)}$, and O$_3$ serve as the dominant atmospheric oxidants during spring, responsible for aerosol NO$_3^-$ formation. In contrast, a lack of sunlight and sea ice influence is experienced during winter, and mixing between two end-members, H$_2$O$_{(v)}$ and O$_3$, best explains the oxygen isotopic composition of the NO$_3^-$ that is formed. Similar to coastal Antarctic sites, reduced daylight hours and/or increased O$_3$ abundance

in the winter and spring MBL lead to greater O$_3$ influence on NO$_3^-$ formation compared to the summer, when OH oxidation chemistry dominates.

Winter is characterized by very low [NO$_3^-$] concentrations with $\delta^{15}$N signatures that reflect background conditions similar to that of the low-latitude Atlantic Ocean (Morin et al., 2009). Interestingly, despite being collected off the coast of South Africa, the N and O isotopic composition of NO$_3^-$ measured for the first wintertime sample reflects a stratospheric NO$_3^-$ source signal. This is also supported by AMBTs that originate near Antarctica, where stratospheric denitrification is reported to occur (Savarino et al., 2007).

Our observations highlight the potential power of N and O isotopes of nitrate in distinguishing between the various natural NO$_x$ sources that result in NO$_3^-$ formation and constraining formation pathways of aerosol NO$_3^-$. In order to improve the utility of the N and O isotopes in the polar atmosphere, more measurements of the isotopic composition of the regional sources, e.g. snow on sea ice, and regional processes, e.g. OH from HONO and sea ice oxidant emissions, are needed. Even though it is complex, the utility of the N isotopes in distinguishing between the various natural NO$_x$ sources that result in NO$_3^-$ formation in the MBL of the Atlantic Southern Ocean, especially in the less frequently sampled seasons of winter and spring, is evident. Furthermore, the O isotopes were able to help constrain formation pathways of aerosol NO$_3^-$ seasonally. This is especially important in the Atlantic Southern Ocean, where oxidation chemistry is poorly constrained (Hosaynali Beygi et al., 2011). The contribution of sea ice to oxidant production when sunlight returns in spring is also highlighted by the O isotopes. As such, these data may be useful to modelling efforts attempting to characterize N cycling between the surface ocean and lower atmosphere and may help improve atmospheric oxidant budgets that are less understood in unpolluted low-NO$_x$ environments.

**Data availability.** Datasets for this research are available at https://doi.org/10.5281/zenodo.7142722 (Burger et al., 2022b).

**Supplement.** The supplement related to this article is available online at: https://doi.org/10.5194/acp-23-1-2023-supplement.

**Author contributions.** KEA designed the study and sampling campaign, acquired funding, and supervised the research. KEA and MGH provided financial and laboratory resources and assisted in data validation. EJ performed laboratory analysis of samples at Brown University. KAMS and JMB conducted the sampling at sea, and JMB performed laboratory analysis at the University of Cape Town. JMB analysed the data and prepared the manuscript with contributions from all co-authors. KEA, MGH, and EJ assisted with reviewing and editing the manuscript.

**Competing interests.** At least one of the (co-)authors is a member of the editorial board of *Atmospheric Chemistry and Physics*. The peer-review process was guided by an independent editor, and the authors also have no other competing interests to declare.

**Disclaimer.** Publisher's note: Copernicus Publications remains neutral with regard to jurisdictional claims in published maps and institutional affiliations.

**Acknowledgements.** We thank the captain and crew of the R/V *SA Agulhas II* for their support at sea and the Marine Biogeochemistry Laboratory in the Oceanography Department at the University of Cape Town for their assistance in the field and in the laboratory. We thank Ruby Ho for analytical support. We thank Riesna Audh, Raquel Flynn, Shantelle Smith, Eleonora Puccinelli, Sina Wallschuss, Eesaa Harris, and Sive Xokashe for nitrite concentration measurements and Sarah Fawcett and Raquel Flynn for quality-controlling the nitrite concentration data. We thank the South African Weather Service (SAWS) for atmospheric temperature, sea level pressure, and relative humidity data during all three voyages. This project has received funding from the European Union's Horizon 2020 research and innovation programme under grant agreement no. 101003826 via the CRiceS (Climate Relevant interactions and feedbacks: the key role of sea ice and Snow in the polar and global climate system) project.

**Financial support.** This research has been supported by the South African National Research Foundation through a Competitive Support for Rated Researchers Grant to Katye E. Altieri (grant no. 111716) and a South African National Antarctic Programme postgraduate fellowship to Jessica M. Burger and a grant to Katye E. Altieri (grant no. 110732). This research was further supported by the University of Cape Town through a University Research Council launching grant and VC Future Leaders 2030 grant awarded to Katye E. Altieri. Additional support was provided by the National Research Foundation through a doctoral scholarship to Jessica M. Burger (grant no. 138813) as well as by the European Union's Horizon 2020 research and innovation programme (grant no. 101003826) via the CRiceS project. This work was partially supported by the NSF (award no. 1851343) via the North Pacific Atmosphere project grant awarded to Meredith G. Hastings. TS16

**Review statement.** This paper was edited by Eliza Harris and reviewed by two anonymous referees.

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

## Remarks from the language copy-editor

## Remarks from the typesetter