# Peer review of "A seasonal analysis of aerosol NO3- sources and NOx oxidation pathways in the Southern Ocean marine boundary layer"

_Atmospheric Chemistry and Physics, 2022_

## Author Comment (AC1)

We thank the reviewer for their supportive and constructive comments on the manuscript. We feel that the paper has been improved by the review process. Below, we address each of the reviewer's specific and technical lineby-line comments. The reviewer comments are in black text, while the responses are in *blue italics* and new text added to the manuscript is in blue.

**Specific comments from Reviewer 1:**

Line 18: Here in the abstract authors mention that the major NOx source in low latitude region is lightning based on d15N, whereas in line 299-305 they conclude that the possible NOx source includes other natural sources such as biomass burning and soil emission. Biomass burning and soil microbes can supply NOx with d15N of -7 to 12 ‰ (Fibiger and Hastings, 2016) and -60 to -14 ‰ (Miller et al., 2018), respectively. Thus it would be difficult to rule them out from possible sources based solely on d15N. The abstract should be corrected to be consistent with discussion.

The abstract has been edited to be consistent with the discussion of potential sources at the low latitudes as follows: "Based on  $\delta^{15}$ N-NO3-, the dominating primary NOx sources were likely a combination of lightning, biomass burning and/or soil emissions at the low latitudes, as well as oceanic alkyl nitrates and snowpack emissions at the mid and high latitudes, respectively."

Line 94-95: Theoretical mechanistic of non-mass dependent isotope signature in ozone is thought to originate from the stabilization step of asymmetric molecules of excited ozone (O3\*), as mentioned in Ireland et al. (2020) and initially proposed by Heidenreich and Thiemens (1986). It is not believed to be associated with photochemistry. Please revise the explanation accordingly.

The explanation of the origin of the non-mass dependent isotope signature in ozone has been revised as follows: "Non-mass dependent fractionation occurs in the troposphere and is thought to originate from asymmetric molecules of excited ozone (O3\*) that lose excess energy via stabilisation to product O3 (Heidenreich & Thiemens, 1986; Ireland et al., 2020)."

Line 103: "a lack of exchange of O atoms with O3" is not correct expression because formation of nitrate is not equilibrium reaction, unlike the case of isotopic exchange between H2O and OH. I suggest rephrasing to "increased contribution from other oxidants".

A lack of exchange of O atoms with O3, has been rephrased to "increased contribution from other

**oxidants".**

Line 166: The authors report that field blanks represented 32% and 59% of the NO3- on sample filters in winter and spring respectively. While they corrected concentration measurements for these field blanks, there is no mention about corrections of isotopic measurements. To mitigate the potential impact of blanks, it is common to collect an excessive amount of nitrate relative to the blank on each filter, or to measure isotopic compositions of the blanks to correct those of samples (e.g., Savarino et al., 2007). But in this manuscript, authors conducted ~24 hours sampling to obtain higher temporal resolution, which resulted in small nitrate loadings on the filters. What is source of the field blanks, and would it be possible to discount the significant impact on isotopic signatures? Is it possible to assume isotopic signatures of the blanks? Even if not, the potential impact and the assumption made for the later interpretation should be carefully addressed.

The amount of nitrate on each stage of the field blank filters is roughly similar (e.g., average of 214 and standard deviation of 41 nmols per filter across stages 1 through 4). The spring and winter sample concentrations are much lower than the summer concentrations, therefore the percentage contribution of the blank to the total signal is larger in spring and winter. In order to facilitate a seasonal comparison, it was important to not increase the number of sampling hours too much from summer to winter to spring. The blank extract concentrations were all less than 1.5  $\mu$ M, therefore we did not have enough volume to measure the isotopic composition directly.

In evaluating the potential sources of the blank, we concluded that it was unlikely to have a vastly different  $\delta^{15}N$  than the sample nitrate for several reasons. First, and importantly, in the figures below the percent contribution of the blank vs.  $\delta^{15}N$ - and  $\delta^{18}O$ -NO3- for spring and winter show no significant relationship indicating that the measured signal is not being driven by a blank. Second, the sodium and chloride values are not unusually high in the blank filters, which lead us to conclude that there was no contamination with seawater. There is also not an unusually high value for sulfate, which makes us confident that ship stack emissions are not the source of the blank.

Finally, the coarse mode  $\delta^{15}N$  is a mass weighted average of stages 1 through 4 for each filter deployment. As a result, samples where the blank is a high proportion of the total signal result in low sample nitrate concentrations, and that stage will then have a relatively low influence on the resulting mass weighted average d15N. If the blank was greater than the sample concentration for a given stage that value was not used in the mass weighted average  $\delta^{15}N$ .

This is now noted in section 2.2.2: "It is important to note that given the low  $[NO_3^-]$  of the field blanks (< 1.5  $\mu$ M), no isotopic analysis could be performed on the blank filters and therefore the blank was not subtracted from the isotope results. However, we note that there was no relationship found between the blank percent contribution and  $\delta^{15}$ N- or  $\delta^{18}$ O-NO3- for spring and winter. This indicates that the measured signal is not driven by the blank contribution."

The blank percentage of sample (Blank %), versus  $\delta^{15}$ N-NO3- and  $\delta^{18}$ O-NO3- in spring (a and b, respectively) and winter (c and d, respectively).

Line 200: Is 72-hour AMBT enough to trace NOx source? I would expect any references or discussion to certify the lifetime of nitrate for 72-hours.

Estimates for the atmospheric lifetime of nitrate in the atmosphere range from about 3 to 5 days (Lu et al., 2021). The spatial uncertainty associated with the location of HYSPLIT generated air mass back trajectories increases the further back in time they are used. It was therefore necessary to use the most conservative time frame, while still ensuring that the lifetime of nitrate in the atmosphere is accounted for. Therefore, a lifetime of 3 days was chosen. In certain cases, i.e., at the lower latitudes near coastal Southern Africa, some 120 day air mass back trajectories were analysed to ensure that even for the upper range of nitrate atmospheric lifetime estimates, no continental influence was experienced near South Africa. In the case of the mid and high latitudes, using 120 hour air mass back trajectories made no difference to the interpretation of the results and thus it was decided to use the most conservative air mass back trajectory path length.

We have now included references in the methods section 2.3 where the nitrate lifetime is stated, and explained why we used the lower estimate of the nitrate lifetime range for plots of air mass history, as follows: "Model estimates of the atmospheric lifetime of  $NO_3^-$  range from approximately three to five days (Lu et al., 2021). AMBTs become increasingly uncertain the further back in time they are used (Sinclair et al., 2013). To minimize this uncertainty, the shortest possible AMBTs are generated while still accounting for the lifetime of  $NO_3^-$  (i.e., 72-hours). Daily 120-hour AMBTs computed for the duration of each voyage were additionally computed (Fig. SX), to confirm that even when utilising the maximum estimate for  $NO_3^-$  atmospheric lifetime, no continental influence from southern Africa is expected."

We have also included an additional supplementary figure which shows 120 hour AMBTs computed for each hour of every voyage, to confirm the lack of influence from continental Southern Africa.

---

## Author Comment (AC2)

We thank the reviewer for their supportive and constructive comments on the manuscript. We feel that the paper has been improved by the review process. Below, we address each of the reviewer's specific and technical line-by-line comments. The reviewer comments are in black text, while the responses are in *blue italics* and new text added to the manuscript is in blue.

GENERAL COMMENTS

The authors present new shipborne measurements during winter and spring of the stable isotopes of nitrogen (d15N) and oxygen (d18O, D17O) isotopes in the coarse mode of atmospheric nitrate collected in the marine boundary layer (MBL) between South Africa and the marginal sea ice zone in Antarctica. d15N values are used to attribute primary sources of atmospheric nitrate: during spring/summer lightning, ocean (alkyl nitrates) and snowpack NOx emissions dominated at low, mid and high latitudes, respectively. During winter transport of NOx precursors such as PAN from lower latitudes as well as potentially stratospheric nitrate contribute mostly to the atmospheric nitrate background. Using D17O and d18O values in an isotope end member mixing analysis the authors confirm the current understanding that oxidation during daytime is dominated by OH and during night time/ winter by O3. They speculate that a third end member emerging at sunrise in spring may be attributed to the onset of halogen chemistry and contribution to oxidation via peroxy radicals.

These are important new atmospheric data from the Southern Ocean MBL covering seasons which are notoriously under-sampled, and therefore should be published. However there are some weaknesses in data interpretation, some gaps in the cited literature as well as presentation of results can be improved.

Major points:

- the introduction should expand on the nitrogen chemistry relevant for the oxygen and nitrogen isotope transfer, i.e. spell out key reactions of the relevant pathways: Step1) NO,NO2 interconversion (fast) and Step2) NO2 oxidation to form nitrate (slower). This will help the reader to follow the arguments presented and assess key uncertainties and missing variables for future studies aiming at a quantitative isotope budget.

*Based on the reviewers suggestion, key reactions and relevant pathways have been included in the introduction as follows, with original text in black and new text in blue:*

*"In addition to there being multiple $NO_x$ sources across the Southern Ocean MBL, several different oxidation pathways can be responsible for $NO_x$ to $NO_3^-$ conversion, varying with chemistry and time of day (Savarino et al., 2007).* Once emitted, NO is rapidly oxidised by ozone ($O_3$) (R1), peroxy radicals ($RO_2$ or $HO_2$) (R2), and/or halogen oxides (XO; where X = Br, Cl, or I) (R3), to $NO_2$.

$NO + O_3 \rightarrow NO_2 + O_2$ (R1)

$NO + RO_2 \text{ (or } HO_2) \rightarrow NO_2 + RO \text{ (or } OH)$ (R2)

$NO + XO \rightarrow NO_2 + X$ (R3)

$NO_2 + O_2 + hv \rightarrow NO + O_3$ (R4)

Under sunlit conditions, $NO_2$ is readily photolyzed to regenerate NO and $O_3$ (R4). The recycling of $NO_x$ between NO and $NO_2$ happens much faster than $NO_x$ oxidation to $NO_3^-$ during the day (Michalski et al., 2003). On a global scale, NO is primarily oxidised to $NO_2$ by $O_3$, followed by $HO_2$ and $RO_2$, while NO to $NO_2$ oxidation via XO is relatively minor (Alexander et al., 2020).

During summer in the Southern Ocean MBL, $NO_2$ is subsequently oxidised primarily by hydroxyl radicals (OH) to form $HNO_3$ (R5).

$NO_2 + OH + M \rightarrow HNO_3 + M$ (R5)

In winter, under dark conditions, when the photolytic production of OH stops, $NO_2$ is oxidised primarily by $O_3$ to form nitrate radicals ($NO_3$) (R6). $NO_3$ can then react with $NO_2$ to form dinitrogen pentoxide ($N_2O_5$) followed by hydrolysis on a wet particle surface to form $HNO_3$ (R7-R8).

$NO_2 + O_3 \rightarrow NO_3 + O_2$ (R6)

$NO_3 + NO_2 + M \leftrightharpoons N_2O_{5(g)} + M$ (R7)

$N_2O_{5(g)} + H_2O_{(l)} + surface \rightarrow 2HNO_{3(aq)}$ (R8)

Alternatively, $HNO_3$ can be formed by the reaction of $NO_3$ with hydrocarbons (HC) (e.g., dimethylsulphide (DMS)) (R9).

$NO_3 + HC \ or \ DMS \rightarrow HNO_3 + products$ (R9)

Lastly, halogen chemistry may result in $NO_3^-$ formation via the production and subsequent hydrolysis of halogen nitrates (R10-R11), as has been suggested for coastal Antarctica in summer (Baugitte et al., 2012).

$XO + NO_2 \rightarrow XNO_3$ (R10)

$XNO_3 + H_2O_{(l)} + surface \rightarrow HNO_{3(aq)} + HOX$ (R11)"

- halogen chemistry in step1) NO,NO2 interconversion and step2) NO2 oxidation to form nitrate with respective implications for the oxygen isotope transfer is currently not considered (Section 3.3) and not included in the oxygen isotope mixing model. However, halogens are important in the MBL particularly near/above sea ice or polar ice caps. There is evidence that halogen chemistry acts as a major NOx sink and source of nitrate via the production and subsequent hydrolysis of XNO3 species as observed in coastal Antarctica in summer (e.g. Bauguitte et all, 2012). Thus increases in D17O (or d18O) in nitrate may reflect increased oxidation by XO during step1 and step2 during daytime (mostly spring), possibly closely linked to local NOx emissions of NOx (e,g, Morin et al., 2012). This is because reaction of halogen radicals X (=Cl,Br,I) with ozone lead to the formation of XO

1) X + O3 --> XO + O2 followed by

2) XO + NO --> X + NO2 (e.g. at 2-3 pptv BrO small impact on D17O in NO2 and NO3-; Savarino, 2016)

3) XO + NO2 + M --> XNO3 + M, XNO3 + H2O --> HNO3 + HOX (efficient transfer of D17O of XO and NO2) at halogen levels of only a few pptv there is considerable impact on NO/NO2 ratios (e.g. Savarino et al.,

2016), NOx lifetime (Bauguitte et all, 2012; Frey et al., 2015) and impact on D17O/d18O in atmospheric nitrate. This needs to be mentioned and included in the discussion on latitudinal gradients of d18O/D17O(NO3-).

*A section pertaining to the potential influence of halogen chemistry during spring and its impact on the oxygen isotopic composition of atmospheric nitrate has now been included in the discussion as follows:*

*"Higher $\delta^{18}$O-NO$_3^-$ values in spring compared to summer may originate from NO$_x$ oxidation by XO. In the Antarctic boundary layer, enhanced levels of BrO occur in spring, over sea ice covered areas (Theys et al., 2011). The production of inorganic bromine has been proposed to be related to frost flowers on thin sea ice (Kaleschke et al., 2004) and blowing of saline snow on sea ice (Yang et al., 2010). Significant interaction with sea ice cover was experienced in spring, particularly at the ice edge transect, which could have promoted NO$_3^-$ formation via the BrO pathway, resulting in increased values of $\delta^{18}$O-NO$_3^-$."*

*Additionally the relevant halogen chemistry reactions have been included in the introduction as discussed above.*

- related to the above: negative correlation between d15N and D17O observed in atmospheric nitrate during Arctic spring (Morin et al., 2012) and in inner Antarctica (e.g. Savarino et al., 2016) indicate that snowpack emissions result in enhanced D17O transfer to nitrate. Possible processes include reactions with XO near halogen sources (sea ice, open ocean) or HONO co-emitted with NOx from the snow pack contributing to the local OH budget (e.g. Legrand et al., 2014; Bond et al., 2023). Correlations between the reported D17O(d18O) and d15N especially during spring need to be analysed to discuss the impact of snow emissions and halogens on the isotope transfer. It seems to me that by overlaying Fig. 3 & 5 there is a noteable anti-correlation between d18O and d15N in the spring ice edge measurements.

*We thank the reviewer for this suggestion, and we do indeed see a strong anti-correlation between $\Delta^{17}$O-NO$_3^-$ and $\delta^{15}$N-NO$_3^-$ for samples collected in spring that experienced large sea ice influence (> 75%) as opposed to open ocean influence, determined using the AMBTs of the samples. As per the reviewer's suggestion, we now include a short discussion about the $\Delta^{17}$O-NO$_3^-$, $\delta^{15}$N-NO$_3^-$ relationship in spring, and invoke HONO and halogen chemistry as potential pathways of enhanced $\Delta^{17}$O-NO$_3^-$ production:*

*"Additionally, a strong anti-correlation (r = -0.86) is observed between $\delta^{15}$N-NO$_3^-$ and $\Delta^{17}$O-NO$_3^-$ for samples collected in spring which experience a greater than 75% sea ice influence, determined based on air mass history. A similar relationship was observed at Dome C during summer (Erbland et al., 2013; Savarino et al., 2016). Previous studies found that the production of enhanced $\Delta^{17}$O-NO$_3^-$ in polar regions is linked to the intensity of NO$_x$ emissions from the snowpack (Moring et al., 2012; Savarino et al., 2016). The correlation between $\Delta^{17}$O-NO$_3^-$ and $\delta^{15}$N-NO$_3^-$ could arise from an increased contribution of HONO photolysis to total OH production, which is co-emitted with NO$_x$ from the snowpack (Grannas et al., 2007), and induces a greater $^{17}$O excess in OH compared to the OH production pathway: O($^1$D) + H$_2$O (Savarino et al., 2016). It could also arise from the coupling of snowpack emissions with reactive halogen chemistry as suggested by Morin et al. (2012). The $\Delta^{17}$O/$\delta^{15}$N relationship presented here for the spring samples with air mass histories that indicate extensive*

influence from snow covered sea ice, suggests that snowpack emissions may lead to enhanced $\Delta^{17}O$ transfer to $NO_3^-$."

[Figure]

Figure X. The relationship between $\Delta^{17}O$-$NO_3^-$ and $\delta^{15}N$-$NO_3^-$ in spring (square symbols). In both panels, samples collected along the ice edge are denoted by the red edge colour, with all other samples collected on the north and southbound legs of the voyage denoted by the green edge colour. The colour bar (blues) indicates the percentage sea ice influence experienced by each filter sample as determined using AMBTs. In panel a, all spring samples are included. In panel b, only samples that experienced a sea ice influence > 75% are included. A straight line (grey) is fitted to the data in panel b. Note the difference in x axis scale between panels.

SPECIFIC COMMENTS

L18 I think you mean "the dominating primary NOx sources"

*This is correct, I have updated the text to indicate the main NOx sources.*

L22-24 is the threshold for when you think O3 oxidation dominates 60 or 70 permil? It does not make sense to have two threshold values or you have to explain why they are different in spring vs summer.

*To clarify, the intention here is not to define a threshold value for $\delta^{18}O$ above which we think $O_3$ oxidation dominates. Instead, we are assuming that given that most of summertime $\delta^{18}O$-$NO_3^-$ values are relatively low (below 70‰), OH oxidation is the dominant $NO_3^-$ formation pathway. Likewise, in winter and spring, given that the majority of $\delta^{18}O$-$NO_3^-$ values are relatively higher (greater than 60‰), additional oxidation pathways must be contributing to $NO_3^-$ formation that involve $O_3$.*

*To avoid any confusion, we have removed these values from the abstract, and simply state that: "Greater values of $\delta^{18}O$-$NO_3^-$ in spring and winter compared to summer, suggest an increased influence of oxidation pathways that incorporate oxygen atoms from $O_3$ into the end product $NO_3^-$ (i.e., $N_2O_5$, DMS and XO)."*

L26-27 not only HO2/RO2 but also oxidation by XO (see related comments)

*The influence of XO in spring has now been included in the abstract as follows:*

*"Significant linear relationships between $\delta^{18}O$ and $\Delta^{17}O$ suggest isotopic mixing between $H_2O_{(v)}$ and $O_3$ in winter, and isotopic mixing between $H_2O_{(v)}$ and $O_3/XO$ in spring with the addition of a third endmember (atmospheric $O_2$) becoming relevant in spring."*

L48 NOx emissions from snow are not considered a primary NO3- source, as this is recycled nitrate from atmospheric deposition (oceanic and lower latitude sources) and no3- produced in snow or coming from the sea ice surface/ ocean. Please clarify.

*We agree with the reviewer that NOx emissions from snow are not a primary source for $NO_3^-$. The use of the word primary in this context is used to describe the main/dominant contributors to nitrate that are natural as opposed to anthropogenic. It is not referring to the nature/phase of the source. We have edited the sentence for clarity: "However, regional budgets of $NO_x$ sources can have a variety of anthropogenic and natural contributors. In the summertime Southern Ocean MBL, natural $NO_x$ sources are the main contributors to atmospheric $NO_3^-$ formation (Morin et al., 2009; Burger et al., 2022)."*

L61-62 or by halogens (see comment above)

*This has been included in the text: "In addition to there being multiple $NO_x$ sources across the Southern Ocean MBL, several different oxidation pathways can be responsible for $NO_x$ to $NO_3^-$ conversion, varying with chemistry and time of day (Savarino et al., 2007). Once emitted, NO is rapidly oxidised by ozone ($O_3$) (R1), peroxy radicals ($RO_2$ or $HO_2$) (R2), and halogen oxides (XO; where X = Br, Cl, or I) (R3), to $NO_2$"*

L67-68 a few pptv of BrO are sufficient. Please expand following above comment.

*We have expanded on this based on the above comment as follow: "Lastly, halogen chemistry may result in $NO_3^-$ formation via the production and subsequent hydrolysis of halogen nitrates (R10-R11), as has been suggested for coastal Antarctica in summer (Baugitte et al., 2012).*

*$XO + NO_2 \rightarrow XNO_3$ (R10)*

*$XNO_3 + H_2O_{(l)} + surface \rightarrow HNO_{3(aq)} + HOX$ (R11)"*

L83 Note that using d15N in nitrate as a source tracer works only if any of the processes involved does not induce any significant isotopic fractionation. Please clarify.

*We assume that isotope fractionation is negligible in our system. Based on a similar comment from Reviewer 1, we outline the basis for this assumption in section 3.2 as follows: "In remote environments where $O_3$ concentrations largely exceed $NO_x$ concentrations, as is the case for the remote Southern Ocean, $NO_x$ isotopic exchange occurs at a much slower rate than the Leighton Cycle reactions. Therefore, little to no equilibrium isotope fractionation is expressed, and the $\delta^{15}N$ of $NO_3^-$ is assumed to reflect the $\delta^{15}N$ of the $NO_x$ source (Walters et al., 2016)."*

L84-85 It is misleading especially for the modellers amongst the readers to cite only a single number for d15N in atmospheric nitrate originating from snow nitrate photolysis. In particular, d15N in the atmospheric nitrate above snow is not constant but changes after polar sunrise as photolytic recycling and isotope fractionation between snow and atmosphere proceed into summer, going from very negative values to near zero. Thus cite

here a range of values observed in spring (when they are still strongly negative) at relevant polar locations were year-round observations are available (e.g. Wagenbach et al., 1998: Neumayer coastal Antarctica, 1986-92; Winton et al., 2020: Dome C East Antarctic Plateau 2009-15)

*We now cite a range of negative values (-50 to -20‰) as per the reviewers suggestion and cite the literature Wagenbach et al., 1998 and Winton et al., 2020. The text has been amended to read: "This is distinct from the snowpack $NO_x$ source, which typically has a very low $\delta^{15}N$ signature (Berhanu et al., 2014; Berhanu et al., 2015) on the order of -50 to -20‰ (Wagenbach et al., 1998; Winton et al., 2020), depending on the degree of snowpack $NO_3^-$ $^{15}N$ enrichment (Shi et al., 2018)."*

L88 what is the uncertainty (standard deviation) of this value?

*The uncertainty of the value (7.6‰) has been included.*

L97 can serve as a proxy (see comment above on halogen chemistry)

*This sentence has been updated to: '$\Delta^{17}O$-$NO_3^-$ therefore can serve as a proxy for the influence of $O_3$ and/or XO during $NO_3^-$ formation (Berhanu et al., 2012)'.*

L101-102 I strongly recommend to summarise in a table assumed isotope ratios for both

d18 and D17O in the discussed end members (O3, OH, RO2/HO2, H2O etc), including

respective uncertainties and references. Place it either here or later in section 3.3 when

prevalent oxidation pathways are discussed.

*A table has now been included in section 3.3, which outlines the assumed oxygen isotope ratios of all the relevant oxidants, or sources of oxygen molecules such as $H_2O$, utilised for the interpretation of our results.*

Table X: A summary of the oxygen isotope ratios ($\delta^{18}O$ and $\Delta^{17}O$) for the end member oxidants and/or oxidant sources ($O_3$, OH, $HO_2/RO_2$ and $H_2O$) utilised in Sect. 3.3.

| Oxidant/source | $\delta^{18}O$ (‰) | References | $\Delta^{17}O$ (‰) | References |
|---|---|---|---|---|
| Terminal $O_3$ | $126.3 \pm 11.9$ | Vicars & Savarino (2014) | $39.3 \pm 2$ | Vicars & Savarino (2014) |
| OH | $-52.7 \pm 2.8^a$ | Walters & Michalski (2016) | ~0 | Michalski et al. (2011) |
| $HO_2/RO_2$ | $23.88 \pm 0.03$ | Barkan & Luz (2005) | ~0 | Michalski et al. (2011) |
| $H_2O$ | $-13.9 \pm 1.4$ | Dar et al. (2020) | ~0 | Michalski et al. (2011) |

[a]The average $\delta^{18}O$-OH was calculated from the equilibrium fractionation between OH and $H_2O_{(v)}$ (Walters & Michalski, 2016) using the observed atmospheric temperature range for winter and spring and the average $\delta^{18}O$-$H_2O$ (Dar et al., 2020).

L103-04 or XO; of course the O in XO originates from O3

*This has been corrected as follows: "As such, a higher $\delta^{18}O$ or $\Delta^{17}O$ for atmospheric $NO_3^-$ reflects the increased influence of $O_3$ and/or XO on $NO_3^-$ formation, while a lower $\delta^{18}O$ or $\Delta^{17}O$ occurs when there is an increased contribution from other oxidants (Hastings et al., 2003; Fang et al., 2011; Altieri et al., 2013)."*

L157-70 were the steps prior to freezing carried out on the ship right after filter exchange? please clarify.

*The filter samples where immediately stored at –20°C on ship once removed from the cascade impactor. Once back in the laboratory at UCT, filter samples were extracted as outlined in the methods. To clarify this, we now write: "Once back on land, filters were extracted using ultra-clean deionised water (DI; 18 MΩ) under a laminar flow cabinet (Air Science)."*

L166 the blanc values are large compared to the ambient values. What are the N and O isotope ratios of the blancs? Were reported sample isotope ratios also corrected for the blanc contribution? This may actually have quite an impact on the reported values if the blanc comes from an isotopically very different pool.

*A similar comment was made by reviewer 1 and our response is as follows:*

*The amount of nitrate on each stage of the field blank filters is roughly similar (e.g., average of 214 and standard deviation of 41 nmols per filter across stages 1 through 4). The spring and winter sample concentrations are much lower than the summer concentrations, therefore the percentage contribution of the blank to the total signal is larger in spring and winter. To facilitate a seasonal comparison, it was important to not increase the number of sampling hours too much from summer to winter to spring. The blank extract concentrations were all less than 1.5 μM, therefore we did not have enough volume to measure the isotopic composition directly.*

*In evaluating the potential sources of the blank, we concluded that it was unlikely to have a vastly different $\delta^{15}N$ than the sample nitrate for several reasons. First, and importantly, in the figures below the percent contribution of the blank vs. $\delta^{15}N$- and $\delta^{18}O$-$NO_3^-$ for spring and winter show no significant relationship indicating that the measured signal is not being driven by a blank. Second, the sodium and chloride values are not unusually high in the blank filters, which lead us to conclude that there was no contamination with seawater. There is also not an unusually high value for sulfate, which makes us confident that ship stack emissions are not the source of the blank.*

*Finally, the coarse mode $\delta^{15}N$ is a mass weighted average of stages 1 through 4 for each filter deployment. As a result, samples where the blank is a high proportion of the total signal result in low sample nitrate concentrations, and that stage will then have a relatively low influence on the resulting mass weighted average d15N. If the blank was greater than the sample concentration for a given stage that value was not used in the mass weighted average $\delta^{15}N$.*

*This is now noted in section 2.2.2: "It is important to note that given the low $[NO_3^-]$ of the field blanks (< 1.5 μM), no isotopic analysis could be performed on the blank filters and therefore the blank was not subtracted from the isotope results. However, we note that there was no relationship found between the blank percent contribution and $\delta^{15}N$- or $\delta^{18}O$-$NO_3^-$ for spring and winter. This indicates that the measured signal is not driven by the blank contribution."*

[Figure]

*The blank percentage of sampled (Blank %), versus δ¹⁵N-NO₃⁻ and δ¹⁸O-NO₃⁻ in spring (a and b, respectively) and winter (c and d, respectively).*

L192-95 Considering the stability of NO2- in solution - When was NO2- measured? Were samples frozen and kept in the dark? please clarify.

*Seawater sampled collected for NO₂⁻ determination where immediately frozen at -20°C and stored in the dark until analysis. Sample analysis was mostly conducted while on board the vessel. This is now included in the text as follows: "Seawater samples for NO₂⁻ determination where immediately frozen at -20°C and stored in dark conditions until analysis. [NO₂⁻] was analysed using the colorimetric method of Grasshof et al. (1983) using a Thermo Scientific Genesys 30 visible spectrophotometer (detection limit of 0.05 µmol L⁻¹). The majority of seawater [NO₂⁻] analysis was conducted while at sea."*

L197-204 Please provide also vertical information on the calculated back trajectories (this is output produced by default in your HYSPLIT runs), e.g. in the figures. Further below you discuss interaction with ocean/ sea ice/ snow surfaces, this applies only when the air mass arriving at the ship location spent time in the boundary layer.

*Figure 1 has been amended to include the vertical information of calculated air mass back trajectories. This has been done by adding a colour bar to each subplot.*

[Figure]

[Figure]

Figure 1. 72-hour AMBTs computed for each hour of every filter deployment made in winter on both the southbound (Winter S) and northbound (Winter N) voyages, in spring on the southbound voyage (Spring S), northbound voyage (Spring N) and ice edge transect (Spring ice edge) and in summer on the southbound (Summer S) and northbound (Summer N) voyages. Red triangles indicate the ships cruise track during each filter deployment. The AMBTs are coloured by height (m) (blue to green colour bar).

L210, 213 interaction with sea ice. See previous comment.

*The relatively low height (< 100 m) of air mass back trajectories confirms the potential for sea ice interaction. This is included in the discussion with reference to figure 1 as follows: "The potential for sea ice influence is supported by the relatively low height (< 100 m) of AMBTs (Fig. 1)."*

L241 cite also other atmospheric nitrate observation in the relevant sector of coastal Antarctica: Halley 2004-05 (Wolff et al., 2008); Neumayer 1986-92 (Wagenbach et al., 1998)

*As per the reviewer's recommendation we have cited the atmospheric nitrate observations in the relevant sector of coastal Antarctica as follows: "Atmospheric [$NO_3^-$] ranging from tens of ng m$^{-3}$ to approximately 100 ng m$^{-3}$ have been observed for the Southern Ocean MBL during late spring (Morin et al., 2009; Shi et al., 2021) and observations from coastal Antarctic sites in the Atlantic sector showed elevated [$NO_3^-$] (~20 to 70 ng m$^{-3}$) in late spring and early summer (Wagenbach et al., 1998; Wolff et al., 2008)."*

L252 highest in early summer - supposedly due to the spring time depletion in stratospheric ozone. Please clarify.

*Yes, increased UV radiation in spring and early summer is attributed to stratospheric ozone depletion and low noon solar zenith angle, this is now included in the text as follows: "Ground-based studies in Antarctica*

demonstrate that UV radiation is highest in spring and early summer, when stratospheric O$_3$ concentrations are at a minimum and the noon solar zenith angle is low (Aun et al., 2020; Lakkala et al., 2020)."

L265-66 Please check vertical information of the corresponding trajectory to support this.

*We have isolated the initial filter deployment in winter and plotted the air mass history of the sample, colour coded by AMBT vertical height. This has now been included as an additional supplementary figure as seen below. This new supplementary figure shows that near the sea ice edge, some AMBTs originated from 300 to 400 m and descend towards the sampling location where most air masses are at < 100 m.*

[Figure]

Figure S1. 72-hour AMBTs computed for each hour of the first filter deployment made in winter on both the southbound (Winter S). Red triangles indicate the ships cruise track during the filter deployment. The AMBTs are coloured by height (m) (blue to green colour bar).

L272-73 Having a combined figure of all isotopes would make it easier to show this (see comment below).

*We have now combined figures 2,3,5 and S3, as seen below.*

[Figure]

Figure 2. The average coarse mode (> 1 μm) atmospheric nitrate concentration [NO$_3^-$] (ng m$^{-3}$) (a), weighted average δ$^{15}$N of atmospheric nitrate (δ$^{15}$N-NO$_3^-$ (‰ vs. N$_2$)) (b), δ$^{18}$O of atmospheric nitrate (δ$^{18}$O-NO$_3^-$ (‰ vs. VSMOW)) (c) and Δ$^{17}$O of atmospheric nitrate (Δ$^{17}$O-NO$_3^-$ (‰)) (d) as a function of latitude (º S). Winter, spring and summer are denoted by blue diamonds, green squares, and orange circles, respectively. For the summer data, where error bars (± 1 SD) are not visible, the standard deviation is smaller than the size of the marker. Spring data are separated into northbound (NB), southbound (SB) and ice edge legs by clear, light grey

and dark grey fills, respectively for panels b-d. Vertical lines indicate the approximate location of the sea ice edge in summer (orange), winter (blue) and spring (green), identified visually using satellite derived sea ice concentration obtained from passive microwave sensors AMSR2 (Advanced Microwave Scanning Radiometer 2; Spreen et al., 2008).

L283 I am surprised, NOx atmospheric lifetimes are considerably shorter than for instance those of PAN (which in turn is admittedly stable at winter temperatures). How can NOx reach Antarctica from lower latitudes? Can you clarify?

*Previous modelling studies suggest that tropospheric transport of $NO_x$ emitted in the mid to low latitudes (i.e., soil emissions, lightning thermal decomposition of peroxyacetylnitrate (PAN) and fossil fuel combustion), contributes to the Antarctic $NO_3^-$ budget in winter (Lee et al., 2014). PAN decomposition has previously been suggested as a NOx source to coastal Antarctica during winter and early spring (Savarino et al., 2007; Jones et al., 2011). This has now been included in the discussion for clarification as follows:* "Previous modelling studies suggest that tropospheric transport of $NO_x$ emitted in the mid to low latitudes (i.e., soil, lightning, thermal decomposition of peroxyacetylnitrate (PAN) and fossil fuel combustion), contributes to the Antarctic $NO_3^-$ budget in winter (Lee et al., 2014). PAN decomposition has previously been suggested as a NOx source to coastal Antarctica during winter and early spring (Savarino et al., 2007; Jones et al., 2011)."

L316 the triple stable isotopic composition ...

*This has been corrected in the text:* "The initial winter sample had a low concentration indicative of the background conditions; however, the triple stable isotopic composition of the sample confirms that it originated from the stratosphere (see sect. 3.2.1)."

L320-25 Cite also relevant Antarctic observations e.g. Wagenbach et al., 1998: Neumayer coastal Antarctica, 1986-92; Winton et al., 2020: Dome C East Antarctic Plateau 2009-15

*The above Antarctic observations have now been cited:* "Springtime $\delta^{15}$N-NO$_3^-$ is also consistent with long-term records of $\delta^{15}$N-NO$_3^-$ measured at coastal Antarctica (Wagenbach et al., 1998) and on the east Antarctic Plateau (Winton et al., 2020), for the same season."

L328 refrence here also Frey et al., 2009; Erbland et al., 2013.

*These references have been added.* "There is a large isotope effect associated with snow NO$_3^-$ photolysis during summer in the Antarctic (Berhanu et al., 2014, 2015; Frey et al., 2009; Erbland et al., 2013), resulting in the emission of low $\delta^{15}$N-NO$_x$ (~ -48‰) to the overlying atmosphere (Savarino et al., 2007; Morin et al., 2009; Shi et al., 2018; Walters et al., 2019)."

L332 I suppose there are no measurements of d15N in nitrate of the snowpack source? this is an important measurement gap to be addressed in the future.

*Currently there are no measurements of $\delta^{15}$N-NO$_3^-$ in snowpack on sea ice in Antarctica that we are aware of.*

*This measurement gap is now addressed in the conclusions as follows:* "Furthermore, the large sea ice extent characteristic of spring highlights the importance of snow-covered sea ice as a NO$_x$ source, in addition to the

well documented summer source from snow covered continental ice (Jones et al., 2001; Walters et al., 2019; Winton et al., 2020). Currently no measurements of $\delta^{15}$N-NO$_3^-$ from snowpack on sea ice exist, which is an important measurement gap that should be addressed in future studies."

L332-34 Please rephrase in light of the non-stationarity of the d15N in atmospheric nitrate from snow emissions (see comment above)

*In order to acknowledge the fact that snow NO$_3^-$ photolysis does not only lead to very low values of $\delta^{15}$N-NO$_x$ but rather a range of values (-50 to -20‰), depending on the degree of NO$_3^-$ loss from the snowpack and subsequent enrichment in the snow and atmosphere, we amended the text as follows:*

"This suggests that the low $\delta^{15}$N-NO$_x$ likely comes from snow nitrate photolysis from the snow on sea ice, before a net loss of NO$_3^-$ from the snowpack leads to any large $^{15}$N enrichment in the snow and subsequently the atmosphere (Shi et al., 2018). We conclude that NO$_x$ as a result of photolysis of snow nitrate on sea ice can explain the relatively low $\delta^{15}$N-NO$_3^-$ observed in samples collected at the high latitudes on the spring southbound leg and during the ice edge transect (Fig. 3 grey filled squares)."

L343-44 limited influence from ... - the caveat is that this depends on the d15N in nitrate of the local snow source. higher d15N in atmospheric nitrate later in spring/ summer can also originate from snowpack emissions, when the source has become increasingly enriched (see comment above). Please balance your conclusion here.

*We have included a caveat to our conclusion here, explaining that higher atmospheric $\delta^{15}$N-NO$_3^-$ values in spring and summer can also originate from snowpack emissions if the local NO$_x$ source becomes enriched in $^{15}$N, as per the reviewer's suggestion. We then go on to explain that a lack of interaction between sampled air masses and sea ice, as indicated by the AMBTs, suggests that a snowpack emission source was unlikely influencing the samples.*

*The text has been edited as follows: "Higher $\delta^{15}$N-NO$_3^-$ values (-22.7 to -1.0‰) were observed during spring for the northbound leg (Fig. 3 open squares; Fig. 4b). The $\delta^{15}$N of atmospheric NO$_3^-$ that originates from snowpack emissions, depends on the $\delta^{15}$N of the local snowpack NO$_x$ source. $^{15}$N enrichment in the snow due to NO$_3^-$ loss, can lead to increased $\delta^{15}$N-NO$_x$ via photolysis, and ultimately higher values of atmospheric $\delta^{15}$N-NO$_3^-$ (Shi et al., 2018). However, the air mass histories of the samples indicate no contact with surrounding sea ice (i.e., the northbound leg; Fig. 3 open squares; Fig. 4b), suggesting that any influence from snowpack NO$_x$ emissions was limited."*

L345-47 I don't understand this sentence. Please rephrase.

*This sentence has been rephrased to: "The NO which originates from nitrite in seawater is thought to limit sea surface RONO$_2$ production. As a result, elevated nitrite concentrations are required for RONO$_2$ production to occur in seawater (Dahl & Saltzman 2008; Dahl et al., 2012)."*

L347 oceanic RONO2 has been long proposed as an important net primary nitrate source to the Antarctic. This should be mentioned.

*We have adjusted the discussion to state that: "Oceanic RONO$_2$ has been long proposed as an important primary NO$_3^-$ source to the Antarctic (Jones et al., 1991). Recent studies have used modelling and isotopic*

approaches to investigate the potential importance of oceanic $RONO_2$ compared to other sources of $NO_3^-$ in the Southern Ocean MBL, particularly in summer (Fisher et al., 2018; Burger et al., 2022)."

L374 reference here some of the earlier literature (Frey et al., 2009; Berhanu 2014, 2015)

*The earlier references suggested have now been added. "We know that $NO_3^-$ photolysis in snow is associated with a large fractionation, leading to the emission of isotopically light $NO_x$ while the remaining $NO_3^-$ pool becomes enriched in $^{15}N$ (eg., Frey et al., 2009; Berhanu et al., 2014;2015; Shi et al., 2018)."*

L379-80 What is the expected time scale (or lifetime) of aerosol nitrate photolysis? If similar to snow nitrate (on the order of weeks), then it may not be relevant compared to the time scales of transport and deposition.

*The formation of $HNO_3$ was thought to be a permanent $NO_x$ sink in the boundary layer, due to slow photolysis rate of gaseous $HNO_3$, in comparison to deposition. However, this view has been challenged by lab and field studies that show that $HNO_3$ adsorbed on particle surfaces is photolyzed at much higher rates than gaseous $HNO_3$ (Ye et al., 2016). Therefore, we refrain from using this argument to discount the potential of aerosol nitrate photolysis as a NOx source. Given that the time scale of aerosol nitrate photolysis may be relevant against deposition, we have softened the language around our concluding statement as follows:*

*"Since neither of the above scenarios matches the observations, the potential influence of aerosol $NO_3^-$ photolysis as a significant $NO_x$ source to the region during our study is unlikely."*

L390 qualitatively - this study is not a quantitative isotope budget

*This has been changed to qualitatively.*

L406 higher d18O in spring possibly also due to oxidation by XO (see above)

*Additional text has been included in the discussion, which refers to the potential for XO oxidation in spring: "Higher $\delta^{18}O$-$NO_3^-$ values in spring compared to summer may originate from $NO_x$ oxidation by XO. In the Antarctic boundary layer, enhanced levels of BrO occur in spring, over sea ice covered areas (Theys et al., 2011). The production of inorganic bromine has been proposed to be related to frost flowers on thin sea ice (Kaleschke et al., 2004) and blowing of saline snow on sea ice (Yang et al., 2010). Significant interaction with sea ice cover was experienced in spring, particularly at the ice edge transect, which could have promoted $NO_3^-$ formation via the BrO pathway, resulting in increased values of $\delta^{18}O$-$NO_3^-$."*

L417 Water vapour is not an oxidant. I am a bit confused here - oxygen isotope transfer from oxidants: O3, OH (O source atmospheric H2O and O(1D) from O3 photolysis), HO2/RO2 (O source atmospheric O2) please clarify, also how is H2O(v) mixing here.

*We thank the reviewer for this comments, and agree that we need to clarify this point in the discussion. $H_2O$ does not act as on oxidant during $NO_x$ to $NO_3^-$ conversion, but rather serves as an oxygen source during the oxidation process (Michalski et al., 2011). Gas phase $H_2O$ can be incorporated into $NO_3^-$, during $NO_2$ reaction with OH (RX). OH exchanges with $H_2O(g)$, such that $\delta^{18}O$-OH depends on the $\delta^{18}O$ of the $H_2O$ vapor it exchanged with.*

*Previous studies that assess the correlation between the oxygen isotopes of $NO_3^-$ in the marine boundary later during summer (Shi et al., 2021), similarly found that a mixing line between $H_2O$ vapor and $O_3$ was the best fit to the observations. They go on to explain that if an equilibrium isotope fractionation of $^{18}O$ between OH and $H_2O$ vapor is excluded (Michalski et al., 2011), such that the $\delta^{18}O$ of OH is close to that of $H_2O$ vapor, then the lower end-member of the mixing line likely results from OH oxidation. In our case it is also likely that oxygen atoms from $O_3$ and $H_2O(v)$ control the oxygen isotopes of $NO_3^-$ in our study, with OH and $N_2O_5$ oxidation being the dominant pathways. In spring, XO oxidation may also lead to high $\delta^{18}O$-$NO_3^-$ and $\Delta^{17}O$-$NO_3^-$, similar to the $N_2O_5$ oxidation pathway. The figure has been updated and labelled appropriately to reflect this. Our data also suggest that a large equilibrium fractionation between OH and $H_2O$ cannot account for the winter and spring observations, which are best explained by $\delta^{18}O$-$OH \sim \delta^{18}O$-$H_2O$. This is now included in the discussion. In addition, given that not all the spring samples are collected at the high latitudes ($60^o$ S to $70^o$ S), for which an additional $H_2O_{(v)}$ end member equivalent to the minimum observed by Dar at al., 2020 (-27.5‰) was included, we now also include average $\delta^{18}O$-$H_2O_{(v)}$ determined between $\sim33^o$ S and $\sim60^o$ S (-13.9 ± 1.4‰). Therefore the figure now includes two $H_2O_{(v)}/ O_3$ mixing lines, both shown in orange. The updated figure is shown below.*

[Figure]

Figure 4. Winter and spring $\delta^{18}O$-$NO_3^-$ vs. $\Delta^{17}O$-$NO_3^-$ are plotted in pane ls (a) and (b), respectively. A straight line (black) is fitted to the data in each panel. In both panels the grey line represents the OH/$O_3$ mixing line, the orange line represents the $H_2O_{(v)}/O_3$ mixing line and the red line represents the $O_2/O_3$ mixing line. In panel b, an additional $H_2O_{(v)}/O_3$ mixing line is included (also in orange) to account for potentially lower values of $\delta^{18}O$-$H_2O_{(v)}$ ($\sim$ -27.5 ‰) at 60 ° to 70 °S.

L414-31 This paragraph will greatly benefit from a table (see comment above on L101-102) to better follow your argument.

*A table has now been included as discussed above.*

L459 increase control of O3 or XO ...

*This has been corrected.*

L492 potentially powerful, but complex (see previous comments); N & O stable isotope measurements of the regional sources (snow, sea ice) are required to achieve a more quantitative budget analysis. consider rephrasing.

*This paragraph was re-phrased as follows:*

*"Our observations highlight the potential power of N and O isotopes of nitrate in distinguishing between the various natural NOx sources that result in $NO_3^-$ formation, and constraining formation pathways of aerosol $NO_3^-$. In order to improve the utility of the N and O isotopes in the polar atmosphere, more measurements of the isotopic composition of the regional sources, e.g., snow on sea ice, and regional processes, e.g., OH from HONO and sea ice oxidant emissions, is needed. Even though it is complex, the utility of the N isotopes in distinguishing between the various natural $NO_x$ sources that result in $NO_3^-$ formation in the MBL of the Atlantic Southern Ocean, especially in the less frequently sampled seasons of winter and spring is evident. Furthermore, the O isotopes were able to help constrain formation pathways of aerosol $NO_3^-$ seasonally. This is especially important in the Atlantic Southern Ocean where oxidation chemistry is poorly constrained (Beygi et al., 2011)."*

TECHNICAL CORRECTIONS

L19 emissions ... originated from ...

*The sentence was modified to indicate where the snowpack emissions come from.* "Based on $\delta^{15}N$-$NO_3^-$, the main $NO_x$ sources were likely a combination of lightning, biomass burning and/or soil emissions at the low latitudes, as well as oceanic alkyl nitrates and snowpack emissions from continental Antarctica or the sea ice at the mid and high latitudes, respectively."

L68 typo: from

*Corrected*

L106 typo: atmospheric

*Corrected*

L107 Antarctic tropospheric oxidation chemistry ...

*Corrected*

L225 In Fig2 I cannot see the second highest winter value of 22 ng/m3, is it covered by

other symbols?

*Yes, this is hidden by the orange circle at the same location. I have re ordered the symbols in the updated figure to make this value easier to see.*

L721 typo: atmospheric

*Corrected*

Figures

Fig1: Label each subplot to help the reader navigate more easily, e.g. 1a. Winter-S 1d. Spring-N ... and include dates in the caption. There is a typo in the caption: ice edge transect should be (e) and N voyage (d)

*The typo in the caption has been corrected, dates of each transect will be included in the caption and each subplot will be given a more descriptive label, as per the reviewer's suggestion.*

Fig2,3 and 5: I strongly recommend to combine these figures including also Fig. S3. This will help to detect a lot more easily common features in [NO3-] and N & O isotope ratios. After all they are related.

*Figures 2, 3, 5 and S3 will be combined into one figure as separate panels.*

To aid interpretation I also suggest to add a panel (or as a separate figure) showing air temperature, radiation (or solar elevation angel) and wind speed at the ship location.

*While solar radiation data is unfortunately unavailable for these cruises we have included a figure of atmospheric temperature and wind speed in the supplementary material as seen below.*

[Figure]

Figure S5. Daily averaged wind speed (a) and air temperature (b) for summer (orange circles), winter (blue diamonds) and spring (green squares), respectively.

Fig.4: Add labels to subplots, e.g. Spring-N ...; 4c: I suspect only the trajectories in bluish colours were within the atmospheric boundary layer above sea ice, whereas the ones with higher d15N (reddish colours) were likely higher up in the free troposphere. This is a point easily supported by including vertical AMBT info (see above).

*More descriptive labels have been added to all subplots.*

REFERENCES

Bauguitte et al., Summertime NOx measurements during the CHABLIS campaign: can source and sink estimates unravel observed diurnal cycles?, Atmos. Chem. Phys., 12(2), pp 989--1002, doi:10.5194/acp-12-989-2012, 2012.

Berhanu et al., Laboratory study of nitrate photolysis in Antarctic snow. II. Isotopic effects and wavelength dependence, J. Chem. Phys., 140(24), doi:10.1063/1.4882899, 2014.

Berhanu et al., Isotopic effects of nitrate photochemistry in snow: a field study at Dome C, Antarctica, Atmos. Chem. Phys., 15(19), pp 11243--11256, doi:10.5194/acp-15-11243-2015, 2015.

Bond et al., 2023, Snowpack nitrate photolysis drives the summertime atmospheric nitrous acid (HONO) budget in coastal Antarctica, Atmos. Chem. Phys. Disc., doi:10.5194/acp-2022-845, 2023.

Erbland et al., Air--snow transfer of nitrate on the East Antarctic Plateau -- Part 1: Isotopic evidence for a photolytically driven dynamic equilibrium in summer, Atmos. Chem. Phys., 13, pp 6403-6419, doi:10.5194/acp-13-6403-2013, 2013.

Frey et al., 2009, Photolysis imprint in the nitrate stable isotope signal in snow and atmosphere of East Antarctica and implications for reactive nitrogen cycling, Atmos. Chem. Phys., doi:10.5194/acp-9-8681-2009, 2009.

Frey et al., Atmospheric nitrogen oxides (NO and NO2) at Dome C, East Antarctica, during the OPALE campaign, Atmos. Chem. Phys., 15(14), pp 7859--7875, doi:10.5194/acp-15-7859-2015, 2015.

Legrand et al., 2014, Large mixing ratios of atmospheric nitrous acid (HONO) at Concordia (East Antarctic Plateau) in summer: a strong source from surface snow?, Atmos. Chem. Phys., 14(18), pp 9963--9976, doi:10.5194/acp-14-9963-2014, 2014.

Morin et al., An isotopic view on the connection between photolytic emissions of NOx from the Arctic snowpack and its oxidation by reactive halogens, J. Geophys. Res., 117, doi:10.1029/2011JD016618, 2012.

Savarino et al., Oxygen isotope mass balance of atmospheric nitrate at Dome C, East Antarctica, during the OPALE campaign, Atmos. Chem. Phys., 16(4), pp 2659--2673, doi:10.5194/acp-16-2659-2016, 2016.

Wagenbach et al., Atmospheric near-surface nitrate at coastal Antarctic sites, J. Geophys. Res., 103(D9), pp 11007--11020, doi:10.1029/97JD03364, 1998.

Winton et al., Deposition, recycling, and archival of nitrate stable isotopes between the air--snow interface: comparison between Dronning Maud Land and Dome C, Antarctica, Atmos. Chem. Phys., 20(9), 5861--5885, doi:10.5194/acp-20-5861-2020, 2020.

Wolff et al., The interpretation of spikes and trends in concentration of nitrate in polar ice cores, based on evidence from snow and atmospheric measurements, Atmos. Chem. Phys., 8(18), pp 5627--5634, 2008.

*Additional references utilised have now been cited.*